# Catalase activity deficiency sensitizes multidrug-resistant *Mycobacterium tuberculosis* to the ATP synthase inhibitor bedaquiline

Boatema Ofori-Anyinam[1,2], Meagan Hamblin[3,12], Miranda L. Coldren[4], Barry Li [1,2], Gautam Mereddy[1,2], Mustafa Shaikh[1,2], Avi Shah[1,2], Courtney Grady[1,5], Navpreet Ranu[3,6,13], Sean Lu[1,2], Paul C. Blainey [3,6,7], Shuyi Ma [4,8,9,10], James J. Collins [3,6,11] & Jason H. Yang [1,2] ✉

Multidrug-resistant tuberculosis (MDR-TB), defined as resistance to the first-line drugs isoniazid and rifampin, is a growing source of global mortality and threatens global control of tuberculosis disease. The diarylquinoline bedaquiline has recently emerged as a highly efficacious drug against MDR-TB and kills *Mycobacterium tuberculosis* by inhibiting mycobacterial ATP synthase. However, the mechanisms underlying bedaquiline's efficacy against MDR-TB remain unknown. Here we investigate bedaquiline hyper-susceptibility in drug-resistant *Mycobacterium tuberculosis* using systems biology approaches. We discovered that MDR clinical isolates are commonly sensitized to bedaquiline. This hypersensitization is caused by several physiological changes induced by deficient catalase activity. These include enhanced accumulation of reactive oxygen species, increased susceptibility to DNA damage, induction of sensitizing transcriptional programs, and metabolic repression of several biosynthetic pathways. In this work we demonstrate how resistance-associated changes in bacterial physiology can mechanistically induce collateral antimicrobial drug sensitivity and reveal druggable vulnerabilities in antimicrobial resistant pathogens.

Despite significant advances in anti-tubercular drug development over the past 80 years, tuberculosis (TB) remains the leading cause of deaths worldwide due to a single bacterial pathogen[1]. *Mycobacterium tuberculosis* (Mtb) is the causative agent of TB disease and standard curative therapy for TB involves at least 4 months treatment with regimen containing the first-line drugs isoniazid (INH) and rifampin (RIF) or rifapentine[2,3]. However, drug-resistant and multidrug-resistant (MDR; defined as resistance to both INH and RIF) TB pose growing threats to global TB control[1]. These pressing challenges have stirred significant drug discovery efforts for curing MDR-TB and have resulted

in drugs such as bedaquiline (BDQ)[4] which inhibits mycobacterial ATP synthase and exhibits exceptional activity against MDR-TB[5]. BDQ is now a cornerstone for several MDR-TB regimens, including bedaquiline-pretomanid-linezolid-moxifloxacin (BPaLM)[6].

Clinical INH resistance is most frequently caused by S315T mutations in the mycobacterial catalase-peroxidase KatG (*Rv1908c*)[7,8]. In INH-susceptible cells, KatG activates the INH prodrug to form an INH-NAD adduct[9] which binds and inhibits the enoyl-acyl carrier protein reductase InhA (*Rv1484*). InhA is essential for the synthesis of mycolic acids that form the mycobacterial cell wall and InhA inhibition is

bactericidal. BDQ targets the C subunit of mycobacterial ATP synthase, encoded by *atpE* (*Rv1305*), and is understood to exert slow mycobacterial lethality by either ATP depletion[10,11] or by electron transport chain uncoupling[12]. Because BDQ is now a cornerstone drug for treating MDR-TB, it is important to understand how and why BDQ is efficacious against MDR-TB to guide future drug discovery efforts.

Here we employed a systems biology approach to investigate mechanisms underlying BDQ hyper-susceptibility in MDR Mtb. We found that BDQ hyper-susceptibility is common in drug-resistant Mtb clinical isolates curated and characterized by the World Health Organization and United Nations[13,14]. We found that deficiencies in catalase activity associated with MDR and non-MDR INH resistance increase BDQ susceptibility. We found that catalase activity deficiency sensitizes drug-resistant Mtb to BDQ by potentiating reactive oxygen species (ROS) formation, by sensitizing cells to DNA damage, by remodeling mycobacterial transcriptional programs, and by metabolically repressing folate biosynthesis. These findings provide mechanistic insight into how physiological changes induced by drug resistance may enable druggable vulnerabilities that may be exploited to cure MDR-TB.

## Results

### Catalase activity deficiency sensitizes MDR *M. tuberculosis* to bedaquiline

BDQ is a cornerstone for treating MDR-TB[6] and KatG loss-of-function is the primary cause of INH resistance[7]. We hypothesized that INH-resistant clinical isolates (and thus MDR clinical isolates) would exhibit increased sensitivity to BDQ. To test this hypothesis, we analyzed BDQ minimum inhibitory concentrations (MICs) from clinical isolates measured by the World Health Organization's Comprehensive Resistance Prediction for Tuberculosis: an International

Consortium (CRyPTIC) consortium[13]. This data includes measurements for 12,289 Mtb clinical isolates from 23 countries in Asia, Africa, South America, and Europe and across 5 Mtb lineages. We found a global reduction in BDQ MICs at the population level for MDR isolates ($n = 3958$) in relation to non-MDR isolates ($n = 7761$) ($p = 3.13 \cdot 10^{-50}$ by Mann–Whitney) (Fig. 1a). Similarly, BDQ MICs were decreased for INH-resistant isolates ($n = 5078$) in relation to INH-susceptible isolates ($n = 5986$) ($p = 8.95 \cdot 10^{-45}$ by Mann–Whitney) (Fig. 1b). 77.9% of these INH-resistant isolates ($n = 3958$) were MDR.

To experimentally validate these findings, we randomly selected 5 INH-resistant and 4 INH-susceptible clinical strains from the World Health Organization- and United Nations-sponsored Special Programme for Research and Training in Tropical Disease *M. tuberculosis* (TDR-TB) strain bank[14]. These strains were randomly selected without first examining information on their genotypes, susceptibilities to antibiotics other than INH, Mtb lineages, or geographical origins (Supplementary Tables 1 and 2). Each INH-susceptible strain possessed INH MICs ≤0.2 µg/mL and each INH-resistant strain possessed INH MICs ≥3.2 µg/mL, as reported by the TDR-TB strain bank. Importantly, these strains were curated in 2012 before the development and use of BDQ as a TB therapeutic and therefore made an ideal set of clinical strains for testing inherent differences in BDQ sensitivity. We treated cells will 2.7 µg/mL BDQ (~20x MIC) and enumerated CFUs after 30 days BDQ treatment. We found that most of INH-resistant clinical strains were significantly more susceptible to BDQ than INH-susceptible clinical strains (Fig. 1c). Importantly, 4 of the 5 INH-resistant isolates were MDR, and 4 of the 5 INH-resistant isolates possessed S315T mutations in KatG, the most clinically prevalent form of INH resistance[7]. These results suggest BDQ hyper-susceptibility in INH-resistant and MDR Mtb cells occurs naturally, is clinically relevant, and may explain BDQ's high efficacy against MDR-TB.

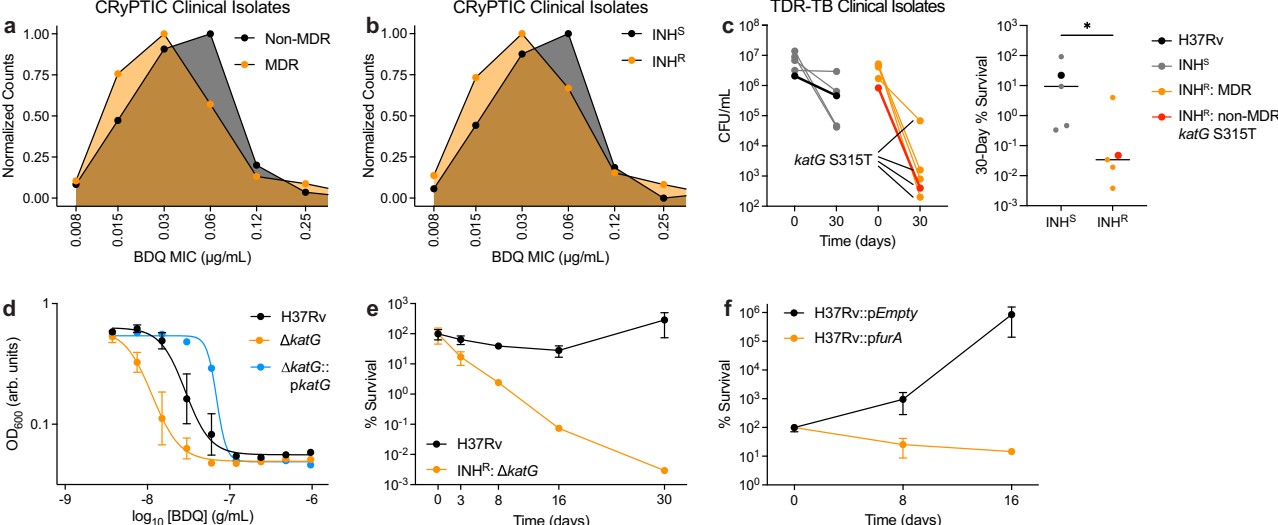

**Fig. 1 | Catalase activity deficiency sensitizes drug-resistant *Mycobacterium tuberculosis* to bedaquiline. a** Normalized count distributions of BDQ MICs for MDR ($n = 3,958$) and non-MDR ($n = 7,761$) clinical strains characterized by the CRyPTIC Consortium[13]. **b** Normalized count distributions of BDQ MICs for INH-susceptible ($n = 5,986$) and INH-resistant ($n = 5,078$) clinical strains characterized by the CRyPTIC Consortium. **c** *Left:* MDR (orange: TDR-TB-19, TDR-TB-31, TDR-TB-193, TDR-TB-198) and non-MDR (red: TDR-TB-42) INH-resistant clinical strains from the TDR-TB strain bank[14] exhibit greater susceptibility to 2.7 µg/mL BDQ than INH-susceptible (gray: TDR-TB-77, TDR-TB-81, TDR-TB-126, TDR-TB-164) clinical strains in 30-day time-kill experiments. Wild-type H37Rv is included for reference (black). *Right:* Relative 30-day survival following BDQ treatment was significantly worse in INH-resistant strains than in INH-susceptible strains as determined by two-sided Mann–Whitney statistical testing ($p = 0.0317$). 4 TDR-TB strains were MDR (TDR-TB-

19, TDR-TB-31, TDR-TB-193, TDR-TB-198). 4 INH-resistant and MDR TDR-TB strains possessed *katG* S315 mutations (TDR-TB-19, TDR-TB-31, TDR-TB-42, TDR-TB-198). $n = 1$ biological replicate for each strain. **d** H37Rv Δ*katG* cells are sensitized to BDQ relative to wild-type cells in 8-day growth inhibition dose-response experiments. *katG* complementation reduces BDQ sensitivity in Δ*katG* cells. $n = 3$ biological replicates for H37Rv and Δ*katG*. $n = 2$ biological replicates for Δ*katG*:p*katG*. **e** H37Rv Δ*katG* cells are hyper-susceptible to 2.7 µg/mL BDQ relative to wild-type cells in 30-day time-kill experiments from $n = 3$ biological replicates. **f** A *furA* over-expression mutant is hyper-susceptible to 2.7 µg/mL BDQ relative to empty vector control cells in 30-day time-kill experiments. Time-kill data quantified as % CFUs normalized to CFUs on Day 0 from $n = 3$ biological replicates. Data depicted as mean ± SEM. Source data are provided in the Source Data file.

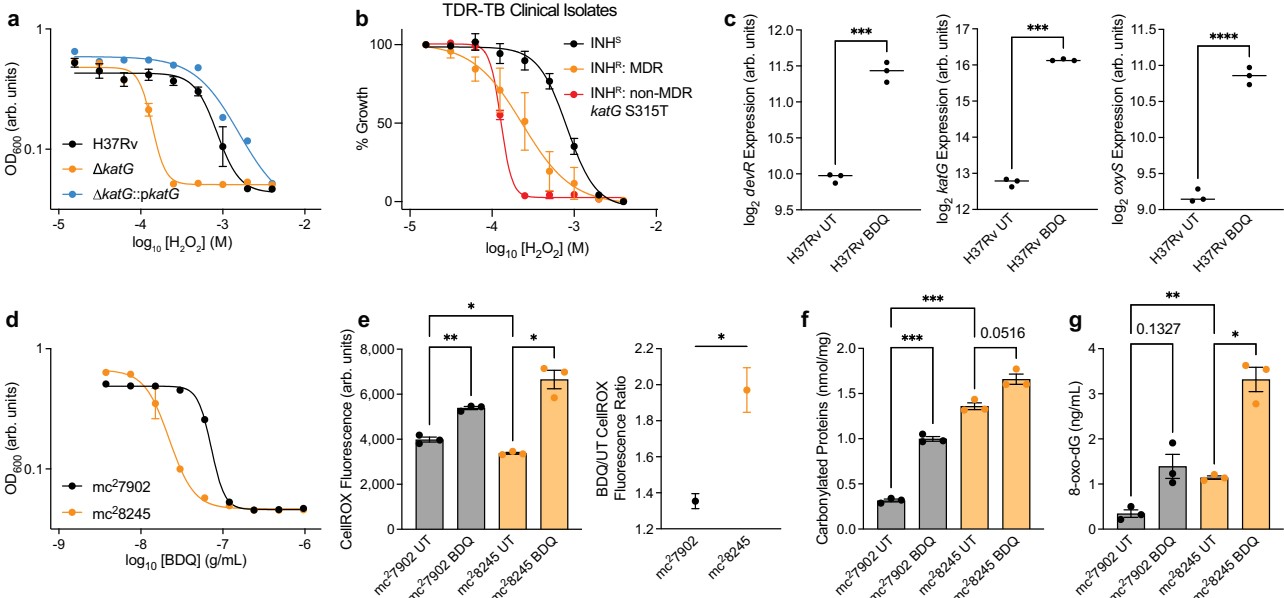

**Fig. 2 | Bedaquiline potentiates reactive oxygen species formation in cells with deficient catalase activity. a** H37Rv Δ*katG* cells are hypersensitive to $H_2O_2$ relative to wild-type cells in 8-day growth inhibition dose-response experiments. *katG* complementation reduces $H_2O_2$ sensitivity in Δ*katG* cells. *n* = 3 biological replicates for H37Rv and Δ*katG*. *n* = 2 biological replicates for Δ*katG*:p*katG*. **b** MDR and non-MDR INH-resistant clinical strains are hypersensitive to $H_2O_2$ relative to INH-susceptible clinical strains in 8-day growth inhibition dose-response experiments. *n* = 4 biological replicates for non-MDR INH-resistant TDR-TB-42 (red). *n* = 1 biological replicate for all other TDR-TB clinical strains. Error bars for INH-susceptible and MDR clinical strains computed across INH-susceptible and MDR clinical strains, respectively. **c** BDQ treatment increases *devR*, *katG*, and *oxyS* expression in wild-type H37Rv cells as measured by RNA sequencing. Expression data reported as smooth quantile normalized $log_2$ sequencing counts. *n* = 3 biological replicates. **d** KatG-deficient mc²8245 cells are hypersensitive to BDQ relative to KatG-replete mc²7902 cells in 8-day growth inhibition dose-response experiments. *n* = 2 biological replicates for mc²7902 and *n* = 3 biological replicates for mc²8245. **e** *Left:* BDQ induces ROS accumulation in mc²7902 and mc²8245 cells as reported by CellROX fluorescence over 4 days treatment with 0.68 µg/mL BDQ. *Right:* ROS accumulation was further enhanced in mc²8245 cells than in mc²7902 cells. *n* = 3 biological replicates. *p* = 0.0279 as determined by two-sided Welch's t-test. Error bars depict standard errors of the mean after error propagation. **f** Protein carbonylation is potentiated in mc²8245 cells relative to mc²7902 cells after 16 hours treatment with 2.7 µg/mL BDQ as reported by ELISA. *n* = 3 biological replicates. **g** Deoxyguanosine oxidation is potentiated in mc²8245 relative to mc²7902 cells after 16 hours treatment with 2.7 µg/mL BDQ as reported by ELISA. *n* = 3 biological replicates. Brown-Forsythe and Welch ANOVA statistical tests were performed on RNA expression, CellROX, protein carbonylation, and deoxyguanosine oxidation experiments, with comparisons between BDQ-treated and untreated cells wild-type and Δ*katG* cells or between untreated wild-type and Δ*katG* as indicated with FDR correction. *$p ≤ 0.05$, **$p ≤ 0.01$, ***$p ≤ 0.001$, ****$p ≤ 0.0001$. Data depicted as mean ± SEM. Source data are provided in the Source Data file.

We hypothesized that deficiencies in KatG catalase activity would directly sensitize Mtb to BDQ. To test this hypothesis, we utilized a Mtb H37Rv *katG* deletion mutant as a model system. Growth inhibition dose-response experiments revealed that deficient catalase activity confers INH resistance (as expected, Supplementary Fig. 1a) and decreases BDQ MIC (Fig. 1d and Supplementary Fig. 1b). Importantly, complementation by constitutive plasmid expression of *katG* rescued wild-type BDQ sensitivity and partially recovered INH susceptibility (Supplementary Fig. 1a). These results support recent observations that laboratory-evolved INH-resistant Mtb auxotrophs can possess decreased BDQ MICs[15]. We additionally performed 30-day BDQ time-kill experiments on wild-type and Δ*katG* cells grown in 7H9 OADC media. Survival of Δ*katG* cells was 4–5 log less than wild-type cells following 30-day treatment with 2.7 µg/mL BDQ (Fig. 1e). These demonstrate that catalase activity deficiency increases BDQ susceptibility in INH-resistant cells.

To further test our hypothesis, we tested BDQ susceptibility in H37Rv cells transformed with a plasmid harboring an anhydrotetracycline (aTc)-inducible *furA* (*Rv1909c*) expression cassette[16–18]. FurA is a member of the ferric uptake regulator (Fur) family of bacterial transcriptional factors and represses *katG* expression in mycobacteria when up-regulated[16,19]. We found that *katG* repression by FurA over-expression resulted in 5-log less CFUs than H37Rv cells expressing an empty vector after 16 days treatment with 2.7 µg/mL BDQ (Fig. 1f). Collectively, these results demonstrate that deficiencies in catalase activity sensitize INH-resistant and MDR Mtb to BDQ.

## Bedaquiline potentiates ROS formation in cells with deficient catalase activity

In most aerobic bacteria, the catalase-peroxidase KatG and alkyl hydroperoxide reductase AhpC mediate protection from oxidative stress by scavenging toxic $H_2O_2$[20]. Expression of these enzymes is normally regulated by the transcription factor OxyR in most bacteria[21]. However, Mtb's *oxyR* homologue possesses several mutations that render it inactive and therefore unable to induce *ahpC* expression in response to oxidative stress[22]. Consequently, KatG is Mtb's primary defense against oxidative stress and KatG-deficient cells are hyper-susceptible to ROS (Fig. 2a).

We hypothesized that INH-resistant and MDR clinical strains harboring a S315T mutation in KatG would be hyper-susceptible to ROS due to deficient catalase activity. We performed $H_2O_2$ growth inhibition dose-response experiments in our TDR-TB clinical strains. Most INH-resistant and MDR clinical strains were hypersensitive to $H_2O_2$ relative to INH-susceptible clinical strains (Fig. 2b and Supplementary Fig. 2a). Interestingly, a non-MDR INH-resistant strain harboring a *katG* S315T mutation (TDR-TB-42) displayed the greatest $H_2O_2$ sensitivity. These results confirmed that catalase activity is commonly deficient in INH-resistant and MDR clinical strains.

To better understand mechanisms underlying how deficient catalase activity sensitizes cells to BDQ, we performed RNA sequencing analyses on wild-type and Δ*katG* cells treated with and without 2.7 µg/mL BDQ for 16 h (Supplementary Data 1). Consistent with previous studies[11,23], BDQ increased expression of the general stress

response regulator *devR* (*dosR*, *Rv3133c*) (Fig. 2c) and several genes involved in central carbon metabolism, including genes in the ATP synthase operon (*Rv1303-Rv1307*). BDQ also decreased expression of ribosomal subunit genes (*Rv0703*, *Rv0719*) in wild-type H37Rv cells. We also found that BDQ induced expression of *katG* and the oxidative stress regulator *oxyS* (*Rv0117*) in wild-type cells (Fig. 2c), suggesting BDQ might also induce ROS formation in Mtb.

BDQ induces oxidative stress responses in non-tuberculous mycobacteria[12] but has never been directly shown to induce ROS accumulation in Mtb[24]. Because KatG is Mtb's primary ROS scavenger, we hypothesized that BDQ might induce higher ROS accumulation in INH-resistant cells than in INH-susceptible cells. To test this hypothesis, we measured BDQ-induced ROS accumulation in the INH-resistant BSL-2 auxotrophic strain mc²8245, which harbors a large genomic deletion spanning over *katG* (H37Rv ΔpanCD ΔleuCD ΔargB Δ2116169-2162530), and its INH-susceptible ancestor mc²7902 (H37Rv ΔpanCD ΔleuCD ΔargB)[25]. We validated that mc²8245 possessed increased sensitivity to BDQ and $H_2O_2$ relative to mc²7902 (Fig. 2d and Supplementary Fig. 2b). We next evaluated BDQ-induced ROS accumulation in these cells using the ROS-sensitive dye CellROX Green. CellROX fluorescence increased following BDQ treatment for both strains indicating BDQ potentiates ROS formation (Fig. 2e). Moreover, the increase in BDQ-induced CellROX fluorescence was significantly greater in mc²8245 cells than in mc²7902 cells. These suggest that BDQ-induced ROS formation is greater in INH-resistant than in INH-susceptible cells.

Respiratory uncoupling has been proposed to underlie BDQ lethality in mycobacteria[12] and is known to potentiate ROS formation. We evaluated the sensitivity of KatG-deficient cells to the respiratory uncouplers carbonyl cyanide m-chlorophenyl hydrazone (CCCP) and nigericin but did not observe differences in CCCP or nigericin sensitivity between Δ*katG* and wild-type cells (Supplementary Fig. 2c). It is therefore unlikely that differences BDQ-induced uncoupling explains BDQ hyper-susceptibility in INH-resistant cells.

We hypothesized that enhanced ROS accumulation in cells deficient for catalase activity might result in increased oxidative cellular damage relative to non-deficient cells. To test this hypothesis, we performed enzyme-linked immunosorbent assays (ELISAs) for protein carbonylation and deoxyguanosine oxidation (8-oxo-dG) in mc²7902 and mc²8245 with or without BDQ treatment. Consistent with the CellROX experiments, protein carbonylation (Fig. 2f) and deoxyguanosine oxidation (Fig. 2g) were greater in mc²8245 cells than in mc²7902 cells following treatment with BDQ. Importantly, protein carbonylation and deoxyguanosine oxidation was also greater in untreated mc²8245 cells than mc²7902 cells, supporting the expectation that catalase-deficient mc²8245 cells would have greater basal oxidative cellular damage than catalase-replete mc²7902 cells. These results indicate that BDQ potentiates ROS formation in Mtb and suggest that enhanced BDQ-induced ROS accumulation may contribute to BDQ hyper-susceptibility in drug-resistant cells.

### Transcriptional programs induced by catalase activity deficiency sensitize Mtb to bedaquiline

We next analyzed transcriptomic differences between wild-type and Δ*katG* H37Rv cells. As would be expected under catalase deficiency, Δ*katG* cells exhibited increased expression of *devR*; the ROS detoxifying enzymes *ahpC*, *ahpD* (*Rv2429*), and *trxC* (*Rv3914*); and antioxidant ergothioneine biosynthesis genes (*Rv3701c-Rv3704c*) (Supplementary Data 1). Interestingly, expression of ATP synthase genes was greater and expression of mycolic acid biosynthesis genes was lower in Δ*katG* cells than in wild-type cells. Moreover, BDQ further suppressed *inhA* expression in Δ*katG* cells than in wild-type cells (Supplementary Fig. 3a). We expected BDQ treatment to further amplify expression of *atpE* (ATP synthase subunit c, the target of BDQ) in Δ*katG* cells than in wild-type cells, but instead found that BDQ

treatment reduced *atpE* expression in Δ*katG* cells to levels similar to untreated wild-type cells (Supplementary Fig. 3b). Because INH inhibition of InhA and BDQ inhibition of ATP synthase are both bactericidal for Mtb, these results suggest that BDQ-induced repression of *inhA* and *atpE* expression may also sensitize cells deficient in catalase activity to BDQ.

We previously found that transcriptional remodeling alters BDQ susceptibility in wild-type Mtb[23]. We hypothesized that altered induction of transcriptional programs in catalase activity-deficient cells might also contribute to BDQ susceptibility. We analyzed differences in transcription factor expression between Δ*katG* and wild-type cells and discovered 12 transcription factors significantly induced at least 2-fold in Δ*katG* cells. These include several transcription factors previously associated with Mtb persistence or drug resistance: *devR*, *prpR* (*Rv1129c*)[26], *blaI* (*Rv2160c*)[27], *mprA* (*Rv0981*)[28], *carD* (*Rv3583c*)[29], *phoY2* (*Rv0821c*)[30], and notably *mmpR5* (*Rv0678*), the gene primarily responsible for BDQ resistance in Mtb clinical strains[31].

To test the hypothesis that induction of these transcriptional programs would sensitize Mtb to BDQ, we performed BDQ growth inhibition dose-response experiments on H37Rv cells over-expressing the three transcription factors with the greatest increases in basal expression in Δ*katG* cells relative to wild-type cells (*Rv3160c*, *kmtR*, *prpR*)[16,17] and an empty vector control (Fig. 3a). We measured the BDQ sensitivity for inhibition of ATP synthesis by BacTiter-Glo and BDQ sensitivity for growth inhibition by turbidity (OD$_{600}$). In support of our hypothesis, BDQ-mediated inhibition of ATP synthesis was increased in cells over-expressing *Rv3160c* or *kmtR* (Fig. 3b, c). In addition, *Rv3160c* over-expression sensitized cells to BDQ-mediated growth inhibition. However, we did not observe changes in BDQ sensitivity for either ATP production or growth in *prpR* over-expressing cells (Supplementary Fig. 3c). Nonetheless, the sensitized BDQ-mediated inhibition of ATP synthesis and growth in *Rv3160c* over-expressing cells suggest that *Rv3160c* and other transcriptional programs induced by deficient catalase activity may also contribute to BDQ hyper-susceptibility in drug-resistant Mtb.

### Catalase activity deficiency sensitizes Mtb to DNA damage

Because BDQ treatment suppressed *atpE* expression in Δ*katG* cells and further suppressed *inhA* expression in Δ*katG* cells relative to wild-type cells (Supplementary Fig. 3a), we hypothesized that other transcriptional changes might also contribute to BDQ hyper-susceptibility in drug-resistant cells. To explore this hypothesis, we hierarchically clustered the RNA expression profiles and defined clusters based on patterns of BDQ-induced gene expression in wild-type and Δ*katG* cells (Supplementary Fig. 4a; Supplementary Data 2). Most transcriptional changes induced by BDQ treatment were shared between wild-type and Δ*katG* cells (Clusters 5 and 6). Gene Ontology analyses revealed BDQ treatment most strongly repressed processes involved in cell wall biosynthesis, including several mycolic acid biosynthesis genes (Cluster 5: BDQ-induced down-regulation in both wild-type and Δ*katG* cells); and potentiated processes involved in cholesterol metabolism (Cluster 6: BDQ-induced up-regulation in both wild-type and Δ*katG* cells). These results are consistent with previous studies in which BDQ-induced changes in H37Rv gene expression were profiled using microarrays[11].

We reasoned that the most informative clusters would contain expression changes that differed between wild-type and Δ*katG* cells. Although Gene Ontology analyses did not reveal any enrichments in Cluster 1 (decreased in wild-type, increased in Δ*katG*) or Cluster 4 (decreased in wild-type, but not in Δ*katG*), several interesting processes emerged from Cluster 2 (increased in wild-type, decreased in Δ*katG*), Cluster 3 (increased in wild-type, but not in Δ*katG*), and Cluster 7 (increased in Δ*katG*, but not in wild-type). Cluster 2 was enriched for ATP synthesis and several stress response pathways. Cluster 3 was enriched for dicarboxylic acid biosynthesis, which includes the

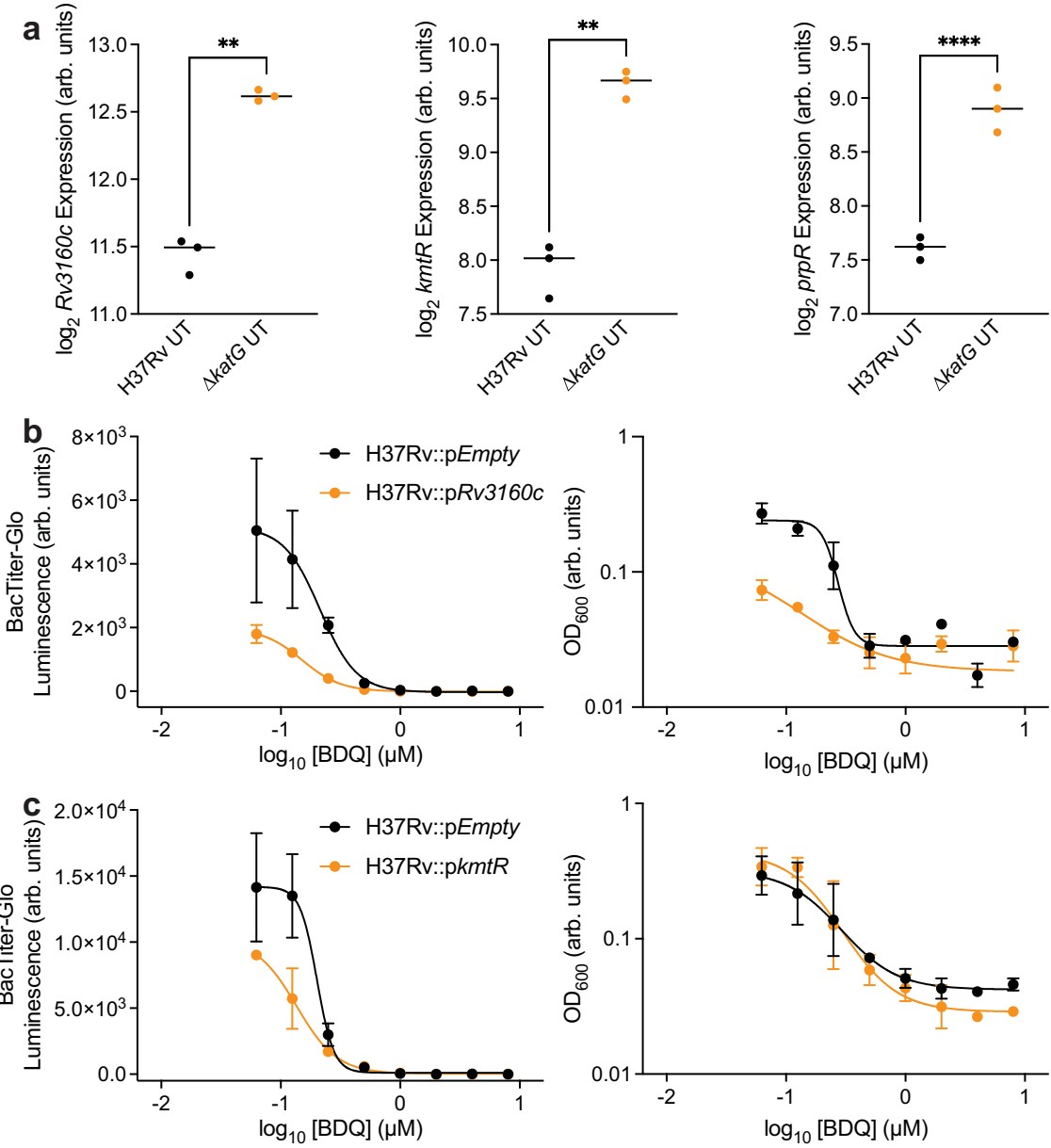

**Fig. 3 | Transcriptional programs induced by catalase activity deficiency sensitize Mtb to bedaquiline. a** Expression of the transcriptional regulators *Rv3160c* ($p = 0.0042$), *kmtR* ($p = 0.0022$), and *prpR* ($p \leq 0.0001$) is higher in untreated Δ*katG* cells than in H37Rv cells as measured by RNA sequencing. Expression data reported as smooth quantile normalized $\log_2$ sequencing counts. Two-sided Welch's t-tests were performed on RNA expression data with comparisons between untreated wild-type and Δ*katG* cells. **b** *Rv3160c* over-expression sensitizes cells to BDQ-inhibited ATP synthesis and growth in 7-day dose-response experiments as determined by BacTiter-Glo and optical density. **c** *kmtR* over-expression sensitizes cells to BDQ-inhibited ATP synthesis in 7-day dose-response experiments. $n = 3$ biological replicates for each experiment. **$p \leq 0.01$. ****$p \leq 0.0001$. Data depicted as mean ± SEM. Source data are provided in the Source Data file.

chorismate and folate biosynthesis pathways, and for tetrapyrrole biosynthesis, which includes much of the Vitamin B$_{12}$ biosynthesis pathway. Cluster 7 was enriched for DNA damage and repair processes.

Consistent with our observation that BDQ increases deoxyguanosine oxidation in Δ*katG* cells relative to wild-type cells (Fig. 2h), we observed significant BDQ-induced increases in the expression of the base-excision repair genes *alkA* and *ung* in Δ*katG* cells over wild-type cells (Fig. 4a). Similarly, the homologous recombination genes *radA* and *recG* were potentiated in Δ*katG* cells over wild-type cells. These are consistent with previous findings that 8-oxo-dG accumulation can enable double-strand DNA breaks following bactericidal antibiotic treatment[32]. These data collectively suggest that deficiencies in catalase activity worsen BDQ-induced DNA damage and sensitize cells to DNA damaging agents. To test this hypothesis, we treated wild-type and Δ*katG* H37Rv cells with the DNA damaging agent phleomycin and observed increased phleomycin sensitivity in Δ*katG* cells relative to wild-type cells (Fig. 4b). Similarly, KatG-deficient mc²8245 cells were hypersensitive to phleomycin relative to KatG-replete mc²7902 cells (Fig. 4c). Importantly, MDR and non-MDR INH-resistant clinical strains were hypersensitive to phleomycin relative to INH-susceptible strains (Fig. 4d and Supplementary Fig. 4b). These results suggest that impaired DNA repair also contributes to BDQ hyper-susceptibility in drug-resistant Mtb.

**Catalase activity deficiency sensitizes Mtb to inhibition of folate biosynthesis**
To further understand how deficient catalase activity alters Mtb's physiology, we performed genome-scale metabolic modeling analyses

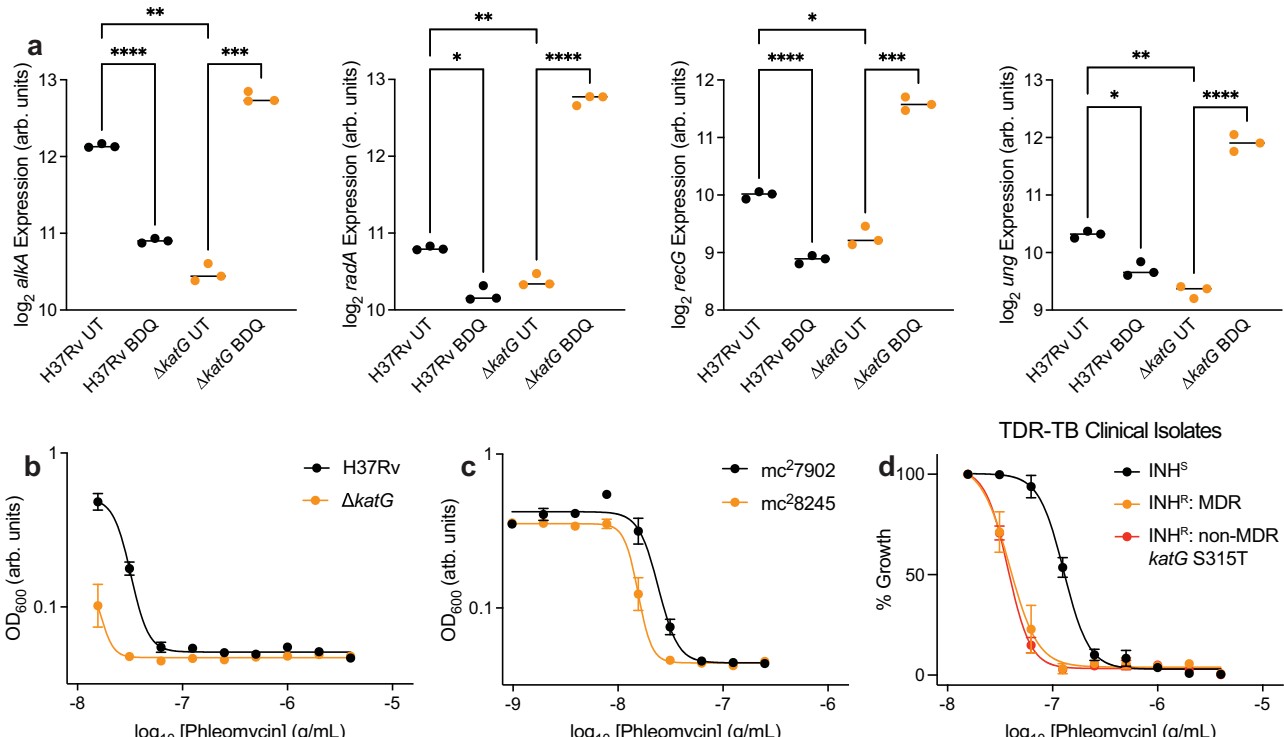

**Fig. 4 | Catalase activity deficiency sensitizes Mtb to DNA damage. a** Expression of several DNA repair enzymes is higher in BDQ-treated Δ*katG* cells relative to BDQ-treated H37Rv cells as measured by RNA sequencing. *alkA*, *radA*, *recG*, and *ung* are members of gene Cluster 7. Expression data reported as smooth quantile normalized log₂ sequencing counts. *n* = 3 biological replicates. **b** Δ*katG* cells are sensitized to the DNA damaging agent phleomycin relative to wild-type cells in 8-day growth inhibition dose-response experiments. *n* = 3 biological replicates. **c** KatG-deficient mc²8245 cells are sensitized to phleomycin in 8-day dose-response experiments. *n* = 3 biological replicates. **d** INH-resistant clinical isolates are sensitized to phleomycin relative to INH-susceptible clinical isolates in 8-day dose-response

experiments. *n* = 4 biological replicates for non-MDR INH-resistant TDR-TB-42 (red). *n* = 1 biological replicate for all other TDR-TB clinical strains. Error bars for INH-susceptible and MDR clinical strains computed across INH-susceptible and MDR clinical strains, respectively. Brown-Forsythe and Welch ANOVA tests were performed on RNA expression data with comparisons between BDQ-treated and untreated cells wild-type and Δ*katG* cells or between untreated wild-type and Δ*katG* with Dunnett's T3 multiple comparisons test FDR correction, as indicated. \**p* ≤ 0.05, \*\**p* ≤ 0.01, \*\*\**p* ≤ 0.001, \*\*\*\**p* ≤ 0.0001. Data depicted as mean ± SEM. Source data are provided in the Source Data file.

using the RNA sequencing transcriptomic profiles as modeling constraints[33]. From the most comprehensive model of Mtb H37Rv metabolism (iEK1011[34]), we generated a Δ*katG*-specific model by removing the catalase reaction (CAT). We then created models corresponding to each of our 4 experimental conditions by applying our RNA-sequencing data as model constraints via the iMAT algorithm[35,36]. We performed flux balance analysis and collected 10,000 flux samples for each metabolic reaction using the optGpSampler algorithm[37] in the COBRA Toolbox[38,39]. These yielded predicted metabolic flux distributions for each metabolic reaction in each sample[33] (Supplementary Data 3).

To verify model fidelity, we first analyzed metabolic reactions related to ROS (Fig. 2) and to processes enriched in the transcriptomic clusters. Consistent with BDQ-induced ROS accumulation (Fig. 2e), model simulations predicted an increase in catalase activity in BDQ-treated over untreated H37Rv cells (Supplementary Fig. 5a). Consistent with enhanced BDQ-induced repression of *inhA* expression in Δ*katG* cells relative to wild-type cells (Supplementary Fig. 3a), model simulations predicted decreased mycolic acid biosynthesis (Supplementary Fig. 5b). Model simulations also predicted decreased activity in propionyl-CoA metabolism (Supplementary Fig. 5c), which also mediates Mtb drug susceptibility[26,40].

Interestingly, the RNA expression profiles revealed significant differences in the expression of folate biosynthesis between Δ*katG* and wild-type cells in both treated and untreated conditions (Cluster 3). BDQ treatment increased expression of *aroF* (Rv2540: chorismate synthase) and *folP1* (Rv3608c: high-affinity dihydropteroate synthase),

but decreased expression of *folP2* (Rv1207: low-affinity dihydropteroate synthase), *folC* (Rv2447c: dihydrofolate synthase), and *dfrA* (Rv2763c: dihydrofolate reductase) relative to wild-type cells (Supplementary Fig. 6a). Consistent with the measured differences in *folC* and *dfrA* expression, metabolic modeling simulations predicted decreased activity throughout the folate biosynthesis pathway (Fig. 5a).

Folate biosynthesis is an essential metabolic process that provides tetrahydrofolate substrates for de novo nucleotide biosynthesis. We recently demonstrated that bactericidal antibiotics increase nucleotide biosynthesis as a homeostatic response to stress-induced nucleotide pool disruptions[41]. Nucleotide metabolism is important for DNA replication and repair, which would be expected to increase under accumulated oxidative DNA damage (Fig. 2g) and increased DNA repair (Cluster 7; Fig. 4a). Consistent with these results, model simulations predicted that BDQ treatment further suppresses nucleotide metabolism in Δ*katG* cells relative to wild-type cells, including downregulation of PRPP synthase and purine and pyrimidine biosynthesis reactions (Supplementary Fig. 6b).

Because model simulations predicted decreased folate biosynthesis in Δ*katG* cells but not wild-type cells (Fig. 5a), we hypothesized cells deficient for catalase activity would be hypersensitive to folate biosynthesis inhibition. To test this hypothesis, we measured the sensitivity of cells deficient for catalase activity and their respective control cells to the antifolates trimethoprim (TMP) and sulfamethoxazole (SMX) in growth inhibition dose-response experiments. Dihydrofolate reductase is the target for TMP and dihydropteroate

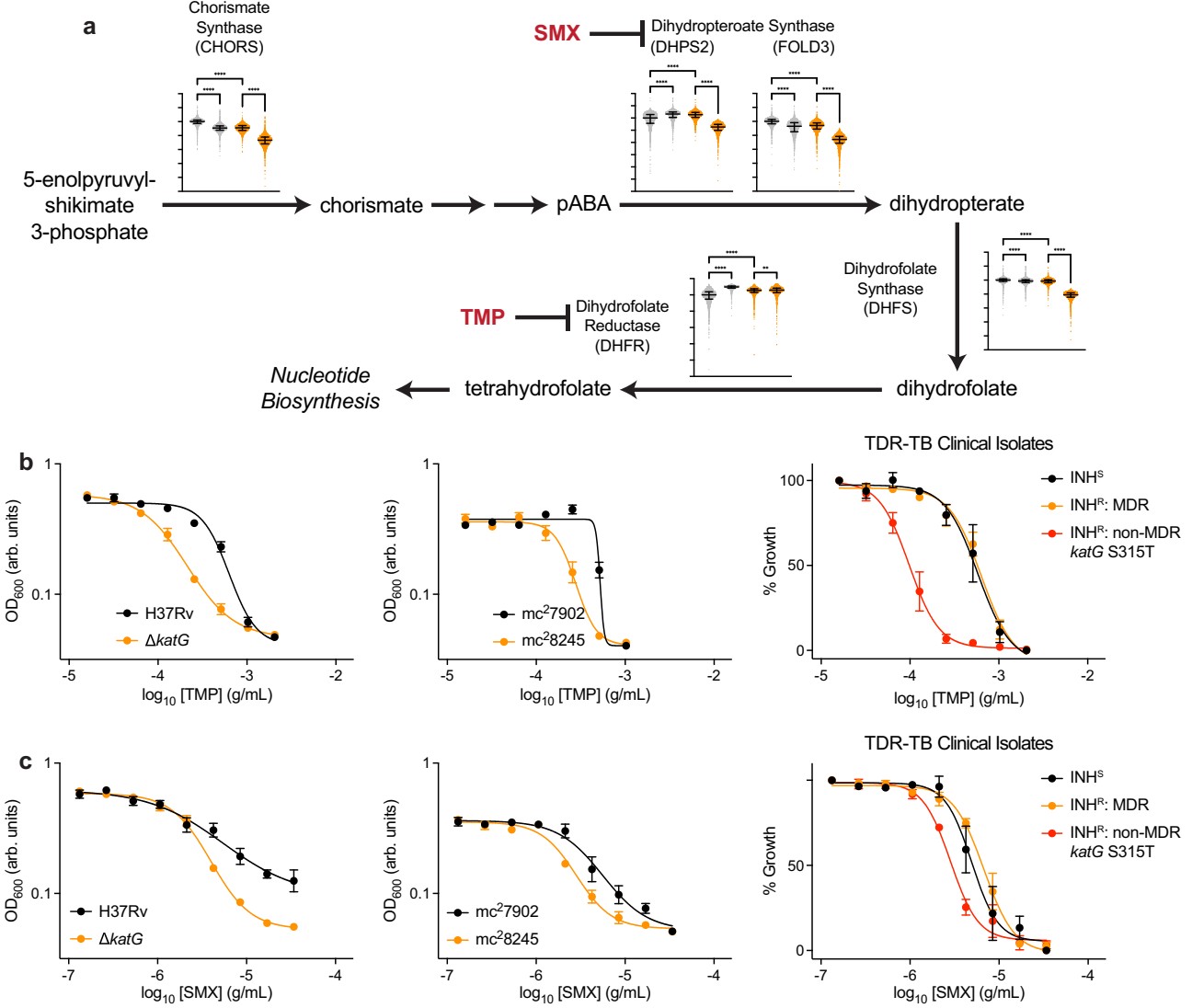

**Fig. 5 | Catalase activity deficiency sensitizes Mtb to inhibition of folate biosynthesis. a** Simulated chorismate synthase (CHORS reaction), dihydropteroate synthase (DHPS2 and FOLD3), dihydrofolate synthase (DHFS), and dihydrofolate reductase (DHFR) activities for BDQ-treated and untreated wild-type and Δ*katG* cells from the iEK1011 Mtb genome-scale metabolic model. Reactions involved in folate biosynthesis are significantly decreased in BDQ-treated Δ*katG* cells relative to BDQ-treated wild-type cells. **b** Catalase activity-deficient Δ*katG*, mc²8245, and non-MDR INH-resistant TDR-TB-42 cells are sensitized to TMP relative to their wild-type, mc²7902, and INH-susceptible controls in 8-day growth inhibition dose-response experiments. MDR clinical strains were not sensitized to folate biosynthesis inhibition relative to INH-susceptible clinical strains. **c** Catalase activity-deficient Δ*katG*, mc²8245, and non-MDR INH-resistant TDR-TB-42 cells are sensitized to SMX relative to their wild-type, mc²7902, and INH-susceptible controls in

8-day growth inhibition dose-response experiments. MDR clinical strains were not sensitized to folate biosynthesis inhibition relative to INH-susceptible clinical strains. $n = 10,000$ flux samples were collected for each metabolic simulation. $n = 3$ biological replicates for experiments involving wild-type, Δ*katG*, mc²7902, and mc²8245 cells. $n = 4$ biological replicates for experiments involving non-MDR INH-resistant TDR-TB-42. $n = 1$ biological replicate for all other TDR-TB clinical strains. Error bars for INH-susceptible and MDR clinical strains computed across INH-susceptible and MDR clinical strains, respectively. Brown-Forsythe and Welch ANOVA statistical tests were performed on RNA expression data with comparisons between BDQ-treated and untreated cells wild-type and Δ*katG* cells or between untreated wild-type and Δ*katG* as indicated with FDR correction. **$p \leq 0.01$, ****$p \leq 0.0001$. Data depicted as mean ± SEM. Source data are provided in the Source Data file.

synthase is the target for SMX. Both enzymes are essential steps of the folate biosynthesis pathway. Consistent with our hypothesis, Δ*katG*, mc²8245 cells, and a non-MDR INH-resistant clinical strain were all hypersensitive to TMP relative to wild-type, mc²7902, and INH-susceptible clinical strains, respectively (Fig. 5b and Supplementary Fig. 6c). Similarly, each of these cells were also sensitized to SMX relative to their respective controls (Fig. 5c and Supplementary Fig. 6c). Interestingly, although the non-MDR *katG S315T* INH-resistant clinical strain (TDR-TB-42) exhibited increased TMP and SMX sensitivity relative to INH-susceptible strains, MDR clinical strains did not exhibit changes in TMP or SMX sensitivity. These suggest that the

mutations in *rpoB* necessary for conferring rifampicin resistance (and thus making INH-resistant strains MDR) compensate for BDQ-induced defects in folate biosynthesis in catalase-deficient cells (Fig. 5a). Moreover, while Δ*katG* and wild-type cells were both sensitized to the combination of TMP and SMX (Supplementary Fig. 6d) relative to TMP alone (Fig. 5b), Δ*katG* cells did not exhibit any further TMP-SMX sensitization than wild-type cells, possibly due to saturation of the growth inhibition that can be achieved by folate biosynthesis inhibition. Nonetheless, these data collectively suggest that BDQ-induced defects in folate biosynthesis may also contribute to BDQ hyper-susceptibility in drug-resistant Mtb.

Collectively, our findings demonstrate that deficient catalase activity causes BDQ hyper-susceptibility in MDR and non-MDR INH-resistant cells through several physiological mechanisms. These include potentiated ROS formation, impaired DNA repair, sensitizing transcriptional remodeling, and inhibited folate biosynthesis.

## Discussion

Here we show that BDQ hyper-susceptibility is common in MDR clinical isolates curated by World Health Organization- and United Nations-sponsored programs (including strains containing *katG* S315T mutations). We report that BDQ's strong efficacy against MTB-TB may be mechanistically explained by several changes in mycobacterial physiology induced by deficient catalase activity. Our observations in Mtb clinical strains are supported by a recent study in Mtb auxotroph strains that found increased BDQ sensitivity following laboratory-evolved INH resistance[15]. Our work here is the first to investigate the mechanistic basis for BDQ hyper-susceptibility in drug-resistant Mtb.

We propose a model in which deficient catalase activity induces several physiological and transcriptional changes which sensitize drug-resistant Mtb to BDQ. We speculate that none of these physiological alterations alone fully account for BDQ sensitization in KatG-deficient cells, but that BDQ hyper-susceptibility is an emergent and epistatic consequence of these several effects. For instance, it is possible downregulated folate and nucleotide biosynthesis in BDQ-treated KatG-deficient cells (Fig. 5a) may limit DNA repair of BDQ-induced oxidative DNA damage (Fig. 2h).

In support of this notion, it is interesting to note that although expression of several transcription factors was increased in Δ*katG* cells in relation to wild-type cells (Fig. 3a), not all these transcription factors increased BDQ sensitivity when over-expressed (Fig. 3c and Supplementary Fig. 3c). While it is possible that the lack of sensitization to BDQ-mediated growth inhibition in *kmtR* and *prpR* over-expressing cells was merely caused by insufficient induction of these genes, we think these results suggest that collateral sensitivity to BDQ (or other antimicrobials) in drug-resistant cells occurs via the epistatic interactions of multiple transcriptional programs. For example, because *kmtR* over-expression enhances BDQ-mediated inhibition of ATP synthesis (Fig. 3c) and because *Rv3160c* over-expression enhances BDQ-mediated growth inhibition (Fig. 3b), the combinatorial effect of increased *Rv3160c* and *kmtR* expression may be greater than the effect from either gene alone in drug-resistant cells.

It is well-known bacterial metabolism is important for antimicrobial efficacy[42] and that INH is only effective against metabolically active Mtb cells[7]. While previous studies primarily focused on oxidative phosphorylation[43], our analyses suggest inhibition of metabolic pathways such as folate or mycolic acid biosynthesis (Fig. 5a and Supplementary Fig. 5b) might also contribute to BDQ hyper-susceptibility in drug-resistant Mtb. Although we did not directly quantify mycolic acid biosynthesis rates, it is reasonable to hypothesize that inhibition of mycolic acid biosynthesis by BDQ would phenocopy the direct actions of INH on INH-susceptible cells. In support of this hypothesis, previous meta-analyses reveal frequent co-occurrence of *katG* and *inhA* mutations in clinical Mtb strains[44], suggesting possible selection for mycolic acid biosynthesis gain-of-function mutants to compensate for naturally occurring mycolic acid biosynthesis defects in KatG-deficient cells.

We propose a conceptual model in which BDQ-induced inhibition of several biosynthetic pathways, including chorismate, folate, propionyl-CoA, and nucleotides, contributes to BDQ hyper-susceptibility. Our work here is extensively supported by a recent study utilizing CRISPRi to study physiological vulnerabilities in laboratory-evolved INH-resistant Mtb auxotroph cells possessing mutations in *katG*[45]. Similar to our findings here, the authors report that *katG* mutant cells are highly sensitive to knock-down of genes involved in oxidative stress response, DNA repair, transcriptional regulators, folate

biosynthesis, chorismate biosynthesis, nucleotide biosynthesis, and mycolic acid biosynthesis. This study orthogonally validates our findings here in Mtb clinical strains derived from the RNA sequencing and genome-scale metabolic modeling analyses.

However, there exists some nuance to our findings. First, although *katG* loss-of-function mutations are the most significant source of INH resistance, mutations in the promoter region of *inhA* and within other genes also confer INH resistance[7]. In this study, we specifically focused on deficiencies in catalase activity caused by *katG* deletion or loss-of-function and did not specifically study other forms of INH resistance. However, we did find increased sensitivity to BDQ (Fig. 1c) and increased susceptibility to DNA damage (Fig. 4d and Supplementary Fig. 4b) in an INH-resistant TDR-TB clinical isolate possessing an *inhA* promoter mutation (TDR-TB-0193; Supplementary Table 2). It remains to be investigated how mutations in the *inhA* promoter may also alter Mtb's physiology.

Moreover, INH-resistance conferring mutations are differentially enriched in different Mtb lineages and it is possible that lineage-specific differences in Mtb physiology may also mechanistically contribute to BDQ susceptibility. For example, *katG* S315T mutations are more prevalent in Lineage 4 strains while *inhA* promoter mutations are more prevalent in Lineage 1 strains[46]. It will be interesting for future studies to uncover how such differences may more broadly impact Mtb susceptibility to other drugs.

Third, our analyses of CRyPTIC drug susceptibility data revealed only modest decreases in BDQ MIC for INH-resistant clinical strains relative to INH-susceptible strains at the population level (Fig. 1b). Although the clinical impact of small differences in BDQ MICs between INH-susceptible and drug-resistant clinical strains is not yet clear, our results demonstrate that BDQ lethality in catalase-deficient cells can be significant (Fig. 1e) despite small changes in MIC (Fig. 1d). Relatedly, recent findings demonstrate that sub-breakpoint differences in INH and RIF MIC are sufficient for predicting TB reinfection after 6-months first-line chemotherapy[47]. These other results highlight how important insights into drug-susceptibility and/or drug-resistance may be overlooked by conventional approaches only measuring MICs.

In addition, although our non-MDR INH-resistant strain containing a *katG* S315T mutation (TDR-TB-42) was sensitized to TMP and SMX relative to INH-susceptible clinical strains, we did not find TMP or SMX sensitization in MDR strains containing *katG* mutations (Fig. 5d and Supplementary Fig. 6c). It is possible that the differences between MDR and non-MDR INH-resistant clinical strains are due to the additional changes in Mtb physiology caused by the *rpoB* mutations that confer RIF resistance. It is also possible that other genomic mutations naturally enriched in INH-resistant clinical strains (such as over-expression of *ahpC*[48]) may suppress antifolate hypersensitivity by compensating for deficient catalase activity. Our results here underscore the need for more mechanistic work to better understand collateral sensitivity mechanisms in MDR-TB.

It is important to note that although relative SMX sensitivity increased in the non-MDR INH-resistant clinical strain relative to INH-susceptible strains, full growth inhibition was not achieved in this INH-resistant clinical strain even at high SMX concentrations (Fig. 5c and Supplementary Fig. 6c). Because Δ*katG*, mc²8245, and the non-MDR INH-resistant clinical strain all exhibited enhanced TMP sensitivity (Fig. 5b), our results suggest dihydropteroate synthase inhibition may exert weaker growth inhibitory effects effect than dihydrofolate reductase inhibition. This weaker effect may due to the fact that Mtb encodes two genes (*folP1* and *folP2*) that can perform dihydropteroate synthesis and the possibility that the expression of these genes may compensate for biochemical inhibition. While TMP and SMX are widely used as bacteriostatic antibiotics against several bacterial pathogens, they are not used in treating TB. Instead, the antifolate para-aminosalicylic acid (PAS) has been used as a

second-line antibiotic for treating MDR-TB[49,50]. However, clinical utilization of PAS for treating (MDR-)TB is limited by its high toxicity[51]. Our data suggest folate biosynthesis may still be useful as a therapeutic target for curing TB. It will be interesting for future studies to mechanistically determine if antifolates are effective against drug-resistant Mtb.

Because BDQ treatment induces oxidative stress regulons in mycobacteria[12] and triggers potentiated Mtb respiration[24], BDQ has been proposed to stimulate ROS production, which participates in antimicrobial lethality in Mtb and other bacteria[52–54]. However, BDQ-induced ROS formation has not previously been directly measured in Mtb and, in contrast, has been reported to not occur at all[24]. Here we find BDQ treatment indeed induces ROS formation in Mtb and that this is potentiated in KatG-deficient cells (Fig. 2e). The discrepancies between our findings here and previous studies may be explained by differences in experimental procedures. ROS are short-lived and highly sensitive to experimental conditions. In the experiments where BDQ-induced ROS were not observed[24], Mtb cells were treated for 1 h and incubated with ROS-sensitive dyes for 30 min before measurement. It is likely that these treatment and incubation durations were insufficient for sensitively detecting changes in ROS accumulation. Here we treated Mtb cells with BDQ for 4 days (during which time BDQ's actions are bacteriostatic) and incubated the cells with CellROX for 24 h and found BDQ-induced ROS accumulation.

Moreover, many ROS-sensitive dyes possess proprietary chemical structures which limit understanding into how these dyes are activated and with what specificity. There is consensus amongst the redox research community that multiple orthogonal assays should be performed to confidently assess ROS formation and oxidative cellular damage[55]. Here we assayed ROS and oxidative damage using 3 independent assays: ROS-sensitive dye oxidation, protein carbonylation, and deoxyguanosine oxidation (Fig. 2e–g). These experiments each support the hypothesis that BDQ sufficiently induces ROS formation in Mtb and conclusively demonstrate that BDQ treatment over a time period relevant to BDQ's slow activity[11] generates measurable ROS.

Our work here also highlights how systems biology approaches, including predictive genome-scale metabolic modeling[33,56], can support mycobacterial research and enable experimentally testable hypotheses[41]. It is important to note that our metabolic modeling simulations predicted decreased folate biosynthesis despite increased gene expression for several genes early in the folate biosynthesis pathway (Fig. 5a and Supplementary Fig. 6a). These suggest that folate biosynthesis is substrate-limited rather than enzyme-limited and demonstrate how analyzing gene expression changes alone may yield misleading interpretations about cell physiology. Efforts integrating transcriptomic[16,17,57], metabolomic[58], fluxomic[59], lipidomic[60], chemogenomic[18,45,61–63], and interpretable machine learning[41,64,65] approaches are key to better understanding how Mtb physiology constrains intracellular infection and drug efficacy[66–69].

Finally, we demonstrated that BDQ hyper-susceptibility is common in MDR clinical isolates, establishing the clinical relevance of our microbiological discoveries. While our work here specifically focused on ATP synthase inhibition, it remains to be studied if MDR Mtb cells might also be hyper-susceptible to other inhibitors of oxidative phosphorylation through similar mechanisms. Understanding the physiological changes induced by MDR is important for rationally designing future MDR-TB treatment regimen[70,71] which may target physiological processes instead of only essential genes[61–63]. For example, while there is considerable enthusiasm in developing new drugs for targeting energy metabolism[69], our results here suggest that drug regimen targeting Mtb DNA repair pathways or inducing sensitizing transcriptional programs may also synergize to treat MDR-TB. These strategies are largely unexplored but provide exciting opportunities for global control of TB disease.

## Methods

### Bacterial strains, growth conditions, reagents

Bacterial strains used in this study were wild-type *M. tuberculosis* H37Rv, a H37Rv KatG deletion mutant, recombinant H37Rv strains transformed with plasmids constitutively expressing *katG* or expressing the transcription factors *furA*, *kmtR*, *prpR* or *Rv3160c* under a tetracycline-inducible promoter[16–18], the Mtb auxotrophic strains mc²7902 (H37Rv Δ*panCD* Δ*leuCD* Δ*argB*) and mc²8245 (Δ*panCD* Δ*leuCD* Δ*argB* Δ*2116169-2162530*)[25], and Mtb strains from the TDR-TB strain bank[14]. All strains were cultured in Middlebrook 7H9 liquid media (Difco) supplemented with 10% oleic acid-albumin-dextrose-catalase (OADC; Difco) and 0.05% Tyloxapol (Millipore Sigma) at 37 °C with shaking. Liquid cultures were sub-cultured on Middlebrook 7H10 solid media supplemented with 10% OADC and 0.2% glycerol (Acros Organics). Liquid and solid media for Mtb auxotrophic strains were additionally supplemented with 24 μg/mL L-pantothenate (Millipore Sigma), 50 μg/mL L-leucine (Millipore Sigma), 200 μg/mL L-arginine (Millipore Sigma), and 50 μg/mL L-methionine (Millipore Sigma). All experiments were performed in at least biological triplicate as indicated.

### Time-kill experiments

Frozen stocks of H37Rv or auxotrophic Mtb cells were inoculated 1:50 into 7H9 culture medium and grown to mid-log phase at an $OD_{600}$ 0.5. Cultures were then back diluted 1:50 into fresh media containing drugs with 0, 0.17, 0.68, or 2.7 μg/mL BDQ (Adooq Bioscience) and/or 0.1 μg/mL INH (Millipore Sigma) as indicated. Cultures were incubated at 37 °C with shaking, sampled at indicated time points, serially diluted in 7H9 media, and plated on 7H10 OADC agar plates. Plates were incubated at 37 °C for 3–6 weeks after which colony forming units were enumerated.

### Drug susceptibility testing

Growth inhibitions experiments were performed on Mtb strains for each drug using the EUCLAST method[72]. Briefly, compound plates containing test drugs were prepared with 2-fold serial dilutions in 7H9 medium. Mtb cells were grown to mid-log phase, diluted 1:100 in 7H9, and 50 μL were dispensed into each well to achieve a final working volume of 100 μL per well. Drugs tested included BDQ, $H_2O_2$, phleomycin, SMX, TMP, CCCP, or nigericin as indicated. TMP-SMX was prepared at a 60:1 ratio as done previously[73]. Plates were incubated for 8–14 days at 37 °C and $OD_{600}$ absorbance was measured on a Biotek Synergy H1 or a BMG Labtech PHERAstar microplate reader at 7, 8, or 14 days as indicated. For experiments involving inducible *furA*, *kmtR*, *prpR* or *Rv3160c* expression, cells were cultured in the presence or absence of 100 ng/mL anhydrotetracycline.

### Reactive Oxygen Species (ROS) quantification

ROS accumulation was measured using the ROS-sensitive dye CellROX Green (Invitrogen) according to the manufacturer's instructions. Briefly, Mtb cells were grown to mid-log and back-diluted to $OD_{600}$ 0.2. 3 mL cultures were incubated for 4 days with and without 0.68 μg/mL BDQ. Cells were dispensed into 96-well plates and incubated with 5 μM CellROX Green for 24 hr at a final working volume of 200 μL per well. CellROX fluorescence was measured on a Biotek Synergy H1 microplate reader at 520 nm.

### Protein carbonylation quantification

Protein carbonylation was measured using the OxiSelect Protein Carbonyl ELISA Kit (Cell Biolabs) according to the manufacturer's instructions. Briefly, Mtb cells were grown to mid-log and cultured with and without 2.7 μg/mL BDQ at a final working volume of 10 mL for 16 hr. Following incubation, cultures were transferred onto ice for 10 min, washed twice with ice cold PBS, and pelleted by centrifugation at $6000 \times g$ for 5 min. 200 μL Bacterial Protein Extraction Reagent

(B-PER II) was added to each pellet with 100 μg/mL lysozyme and 5 U/mL DNase I added. Total protein abundance was quantified for each sample using a Bicinchoninic acid (BCA) assay (Thermo Fisher Scientific) according to the manufacturer's instructions before performing protein carbonylation experiments with the OxiSelect ELISA assay. Absorbance measurements were taken at 450 nm on a Biotek Synergy H1 microplate reader.

## 8-oxo-dG quantification

Protein carbonylation was measured using the OxiSelect 8-OHdG Oxidative DNA Damage ELISA Kit (Cell Biolabs) according to the manufacturer's instructions. Briefly, Mtb cells were grown to mid-log and cultured with and without 2.7 μg/mL BDQ at a final working volume of 10 mL for 16 h. Following incubation, cultures were transferred onto ice for 10 min, washed twice with ice cold PBS, and pelleted by centrifugation at $6000 \times g$ for 5 min. DNA was extracted using the QIAamp DNA Mini purification kit (Qiagen) according to the manufacturer's instructions. Double strand DNA was converted to single strand DNA by incubating samples at 95 °C for 5 min and rapidly chilling on ice. Single strand DNA was then digested before performing 8-OHdG measurements using the OxiSelect ELISA kit. Absorbance measurements were taken at 450 nm on a Biotek Synergy H1 microplate reader.

## RNA sequencing and analyses

Mtb cells were grown to mid-log phase and cultured with and without 2.7 μg/mL BDQ for 16 hr. Immediately following incubation, cultures were pelleted and 1 mL Trizol (Thermo Fisher Scientific) was added to each pellet. Samples were lysed by bead beating and kept at 4 °C for 5 min. Chloroform (Amresco) was added to each sample, lysates were vortexed vigorously, and samples were pelleted by centrifugation at $12,000 \times g$ for 5 min. Supernatants for each sample were carefully transferred to RNase-free tubes for purification by the Direct-Zol RNA Miniprep Plus RNA purification kit (Zymo Research), according to the manufacturer's instructions. Total RNA was eluted in RNase free water and RNA concentrations were measured using a Thermo Fisher Nanodrop One. RNA integrity was confirmed using an Agilent 4200 TapeStation system. Ribosomal RNA was depleted using the Illumina Ribo-Zero Plus rRNA Depletion kit followed by library preparation with the NEBNext Ultra II Directional RNA Library Prep Kit for Illumina (New England Biolabs) as per manufacturer's instructions. RNA sequencing was performed by the Rutgers Genomics Center on an Illumina NovaSeq6000 system with 100x coverage.

Raw sequencing reads were aligned against the NCBI *M. tuberculosis* H37Rv reference genome (NC_000962.3) using *Bowtie 2*[74]. Read counts were compiled using *featureCounts*[75] and quantile normalized by *qsmooth*[76]. Quality data, adapter and quality trimming statistics, and alignment and counts metrics were compiled and assessed using *MultiQC*[77]. Normalized gene counts were hierarchically clustered for clustering analyses.

## Gene set enrichment analyses

Gene set enrichment analyses were performed on Biocyc[78] using the *M. tuberculosis* H37Rv curated database (v. 27.1). SmartTables were created for genes comprising each cluster set as determined by hierarchical clustering. Gene Ontology enrichment terms were identified by performing "Enrichment Analysis" for "GO terms – genes enriched for GO (biological processes) using Fisher Exact statistics and Benjamini-Hochberg corrections for false discovery.

## Genome-scale metabolic modeling

Model simulations were performed using the iEK1011 genome-scale model of *M. tuberculosis* metabolism[34]. RNA sequencing data were applied as modeling constraints using the *iMAT* algorithm[35,36] in the *COBRApy* toolbox[38] with Gurobi Optimizer as the solver. KatG-deficient cells were modeled by setting bounds for the catalase (CAT) reaction to 0 to represent *katG* deletion. Simulations were performed by collecting 10,000 flux samples for each model using *optGpSampler* in the *COBRApy* toolbox[37]. Samples were down-sampled 10-fold for statistical analyses and visualization.

## WHO CRyPTIC MIC data analyses

BDQ minimum inhibitory concentration (MIC) data was downloaded from the WHO CRyPTIC Consortium[13]. Samples with missing BDQ MICs were removed from downstream analysis. BDQ MICs outside the reported concentration rate were replaced with adjacent concentrations on a dilution series as appropriate (MICs "≤0.008" set to 0.008; "≤0.015" set to 0.015; ">1" set to 2; ">2" set to 4). Samples were then segregated to INH-susceptible (INH$^S$) or INH-resistant (INH$^R$) as determined by the CRyPTIC Consortium. Strains were enumerated for each set and MIC and normalized by peak counts for each distribution (INH$^S$ counts normalized to counts at 0.06 μg/mL BDQ; INH$^S$ counts normalized to counts at 0.03 μg/mL BDQ).

## Reporting summary

Further information on research design is available in the Nature Portfolio Reporting Summary linked to this article.

## Data availability

All data are available in the main text, Supplementary Information file, Supplementary Data files, and Source Data file. CRyPTIC Consortium data is available at https://ftp.ebi.ac.uk/pub/databases/cryptic/release_june2022/. RNA sequencing data data generated in this study have been deposited in the Sequence Read Archive database under BioProject accession code PRJNA1139169 [https://www.ncbi.nlm.nih.gov/bioproject/1139169]. Source data are provided with this paper.

## Code availability

Analysis code is deposited on GitHub (https://github.com/jasonhyang/OforiAnyinam_2024) and Zenodo (https://doi.org/10.5281/zenodo.13923984).

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

## Acknowledgements

*Mycobacterium tuberculosis* H37Rv was generously shared by Deborah Hung from the Broad Institute of MIT and Harvard. Plasmids pMSG430 and pMSG430-*katG* were generously shared by Christina Stallings from Washington University in St. Louis. H37Rv Δ*katG*, mc$^2$7902, and mc$^2$8245 were generously shared by William Jacobs from the Albert Einstein College of Medicine. Transcription factor over-expression strains were generously shared by David Sherman from the University of Washington. TDR-TB strains were generously shared by David Alland from Rutgers New Jersey Medical School. The authors thank David Alland, Hassan Safi, Pradeep Kumar, Joel Freundlich, and Jees Sebastian from Rutgers New Jersey Medical School; and David Sherman from the University of Washington for helpful discussions. The authors also thank James Gomez and Zohar Bloom-Ackermann from the Broad Institute of MIT and Harvard for training in Biosafety Level-3 research activities. This work was supported by grants R00-GM118907 (to J.H.Y.), R01-AI146194 (to S.M., J.J.C. and J.H.Y.), U19-AI11276 (to J.J.C.), U19-AI62598 (to S.M. and J.H.Y.), DP2-AI164249 (to S.M.), U24-AI118668 (to P.C.B.), R21-AI121932 (to P.C.B.) and RM1-HG006193 (to P.C.B.) from the National Institutes of Health. N.R. was supported by a NSF GRFP. P.C.B. was additionally supported by the Burroughs Wellcome Fund Career Awards at the Scientific Interface. J.J.C. was additionally supported by HDTRA12210032 from the Defense Threat Reduction Agency and a generous gift from Anita and Josh Bekenstein. J.H.Y. was additionally supported by the Agilent Early Career Professor Award.

## Author contributions

J.H.Y. and J.J.C. conceptualized the study. J.H.Y., J.J.C., S.M. and P.C.B. supervised the project. B.O., M.H., M.L.C., B.L., G.M., M.S., A.S., C.G., N.R., S.L., S.M. and J.H.Y. executed the experiments and data analyses. B.O. and J.H.Y. wrote the manuscript and visualized the data. J.H.Y., J.J.C., S.M. and P.C.B. acquired funding. All authors assisted in manuscript preparation.

## Competing interests

J.J.C. is a co-founder and board member of Phare Bio, a nonprofit venture focused on antibiotic drug development. P.C.B. is a consultant to or holds equity in 10X Genomics, General Automation Lab Technologies/Isolation Bio, Celsius Therapeutics, Next Gen Diagnostics, Cache DNA, Concerto Biosciences, Stately Bio, Ramona Optics, Bifrost Biosystems, and Amber Bio. His laboratory has received research funding from Calico Life Sciences, Merck, and Genentech for unrelated work. None of these interests are connected to this study. The remaining authors declare no competing interests.

## Additional information

[1]Ruy V. Lourenço Center for Emerging and Re-Emerging Pathogens, Rutgers New Jersey Medical School, Newark, NJ 07103, USA. [2]Department of Microbiology, Biochemistry, and Molecular Genetics, Rutgers New Jersey Medical School, Newark, NJ 07103, USA. [3]Infectious Disease and Microbiome Program, Broad Institute of MIT and Harvard, Cambridge, MA 02142, USA. [4]Center for Global Infectious Disease Research, Seattle Children's Research Institute, Seattle, WA 98105, USA. [5]Public Health Research Institute, Rutgers New Jersey Medical School, Newark, NJ 07103, USA. [6]Department of Biological Engineering, Massachusetts Institute of Technology, Cambridge, MA 02139, USA. [7]Koch Institute of Integrative Cancer Research at MIT, Cambridge, MA 02139, USA. [8]Department of Pediatrics, University of Washington, Seattle, WA 98195, USA. [9]Department of Chemical Engineering, University of Washington, Seattle, WA 98195, USA. [10]Pathobiology Graduate Program, Department of Global Health, University of Washington, Seattle, WA 98195, USA. [11]Institute for Medical Engineering and Science, Massachusetts Institute of Technology, Cambridge, MA 02139, USA. [12]Present address: Eversana Consulting, Boston, MA 02120, USA. [13]Present address: insitro, South San Francisco, CA 94080, USA. ✉e-mail: jason.y@rutgers.edu

