## [Peer Review file · Nature Communications]

Catalase activity deficiency sensitizes multidrug-resistant *Mycobacterium tuberculosis* to the ATP synthase inhibitor bedaquiline

Corresponding Author: Professor Jason Yang

Version 0:

Reviewer comments:

Reviewer #1

(Remarks to the Author)

This manuscript by Ofori-Anyinam investigates the impact of isoniazid resistance in *Mycobacterium tuberculosis* strains on the susceptibility towards bedaquiline. BDQ represents a central component of the recently established BPaLM MDR-TB treatment regimen and therefore gaining insight into factors underlying the sensitivity of INH-resistant strains towards BDQ is of key importance.

The authors initially perform an extensive survey of >10,000 *Mycobacterium tuberculosis* clinical isolates and report that INH-resistant strains, compared to INH-sensitive strains, on average display decreased MIC values for BDQ (similar for MDR-strains versus non-MDR strains). Extending previous studies, the authors find that addition of BDQ suppressed the development of resistance against INH and that various KatG mutant strains display enhanced BDQ sensitivity in time kill kinetics. KatG loss of function seems to be the parameter that causes the enhanced BDQ susceptibility in MDR strains.

KatG deficient strains also showed enhanced sensitivity for H₂O₂, but not for uncouplers. In contrast to previous results, the authors found that BDQ induced ROS formation, which was enhanced in KatG deficient strains. This also translated into cellular damage such as increased protein carbonylation and may contribute to the observed BDQ hypersensitivity. Based on RNAseq experiments the authors test if differentially regulated components, such as transcription factors and proteins involved in DNA repair can influence BDQ sensitivity. Indeed, over-expression of *kmtR* and *Rv3160* sensitized the bacteria to BDQ and KatG deficient bacteria, including INH-resistant and MDR clinical isolates, were found more susceptible to DNA damage.

Finally, a genome-scale metabolic model for KatG deficient bacteria was applied to predict metabolic flux changes as compared to wild-type bacteria. The model indicated, in contrast to the RNAseq, decreased folic acid metabolism. The authors experimentally support this prediction and demonstrate increased vulnerability of KatG deficient bacteria for SMX and TMP, two inhibitors of the folic acid biosynthesis pathway (though not for the combination of these two inhibitors). The picture emerges that KatG deficiency is connected to BDQ hyper-susceptibility by enhanced ROS formation, impaired DNA repair, transcriptional remodeling and folic acid biosynthesis inhibition.

These results are important and will have a significant impact on the field.
Please address the following points:

Major points:

1. Figure numbers mentioned in main text in a number of cases do not seem to be accurate, see examples below. This makes navigating the paper difficult.

Line 88-89: Fig 1c meant instead of Fig 2b?

Line 104: Fig 1d instead of Fig 1c?

Line 115: Fig 1e instead of 1 d?

Line 118: 1f instead of 1e?

Line 126: Fig 1i instead of Fig 2h?

Please check throughout the paper.

2. The authors use a variety of KatG deficient strains (large deletion, single gene deletion, point mutation), making it likely that an observed phenotype is really caused by deletion of KatG. Nevertheless, it is recommended to support this conclusion by a complementation study, expressing KatG gene again in the deficient strain. Including such a complementation in one of the presented experiments seems sufficient.

3. Lines 104-106 and elsewhere in manuscript: can the authors describe which measures were taken to minimize drug carry over in the time kill experiments? This seems exceptionally important in case of BDQ, where low concentrations carried along to the agar plate may already prevent bacterial growth.

Minor points:

Line 50: INH-resistant strains typically carry a mutation in KatG. Can the authors provide information if these mutations only prevent the activation of the INH prodrug or if they completely abolish the catalase activity of this enzyme?

Line 125: "reduction of CFU over uninduced cells"

This text probably refers to Fig 1i, however, in that panel bacteria carrying the *pfurA* plasmid are compared to bacteria with empty plasmid. Please clarify. In Fig 1i, the reduction in CFU seems stronger than the described modest 1-2 log.

Line 126-127: something missing in this sentence.

Lines 164-165: "greater increases in BDQ-induced protein carbonylation (Fig. 2f) and deoxyguanosine oxidation (Fig. 2g)" In Fig 2g, the increase upon BDQ addition actually seems lower in 8245 cells as compared to 7902 cells. Please clarify.

Line 173-175: Please confirm that the increased expression mentioned in this sentence was measured in the KatG deficient bacteria.

Line 174-175: "BDQ treatment synergized with *katG* deletion to further suppress *inhA* expression."
How was synergy defined or calculated here?

Lines 202-203: please mention that BDQ was earlier shown to downregulate these pathways (Koul et al 2014, cited elsewhere as ref 11).

Lines 264-265 and Fig 5b: any thoughts why the SMX/TMP combination does not act stronger against the KatG deficient strain as compared to wild-type (while the individual inhibitors do act stronger)?

Reviewer #2

(Remarks to the Author)

Review of NCOMMS-24-00092: Ofori-Anyinam et al., "Catalase deficiency sensitizes multidrug-resistant *Mycobacterium tuberculosis* to the ATP synthase inhibitor bedaquiline."

This manuscript describes studies to understand the mechanistic link between isoniazid resistance and bedaquiline susceptibility in *Mtb*. The authors begin with a re-analysis of the MIC data collected by the CRYPTIC consortium and identify a slight skew in MIC towards more sensitive for bedaquiline in INH-R *Mtb* compared to INH-S *Mtb*. They then recapitulate a slight difference in the BDQ killing kinetics in INH-S vs INH-R strains in another set of a few isolates over 30 days. They spend most of the paper describing studies of laboratory-generated *katG* mutants to tie this effect to oxidative stress, transcription factors, DNA damage and folate metabolism. On balance there may be a slight effect but I remained unconvinced this is clinically meaningful or experimentally proven.

Specific Comments:

(1) Clinically the most abundant *katG* mutant observed is the S315T mutant (estimates range from 70-90% of INH-R isolates are this mutation) which is far from catalase deficient (see PMID: 9291321 among many others). This mutation retains virtually all the catalase function while decreasing the affinity for INH and imparts very little fitness cost to the organism. Although the authors mention this mutant in Figure 1C in the context of the clinical strains, they go on to study laboratory generated *katG* null mutants which is not at all similar to what is seen clinically. Thus, the relevance of the work to the use of BDQ in MDR-TB patients seems a bit of a stretch.

(2) The prevalence of BDQ and INH resistant isolates in the CRYPTIC consortium data set is not straightforward to interpret. As the authors state (lines 93-94), among INH resistant isolates in this dataset, BDQ resistance is less common than is resistance to any of the other antibiotics tested – 1.5% of INH resistant isolates were also BDQ resistant (taken from Table G in the supplementary information of reference 13). However, this may be more a reflection of the fact that BDQ has the lowest

rate of resistance of any antibiotic in this data set (at 0.9%, taken from page 3 of reference 13), rather than a specific relationship between INH and BDQ resistance. A cursory analysis of the numbers reported in reference 13 (including INH resistance rate of 49%, also taken from page 3 of reference 13) suggests that while 1.5% of INH resistant isolates are also resistant to BDQ, only 0.3% of INH susceptible strains are resistant to BDQ. Thus it actually appears that INH resistance is more associated with BDQ resistance, rather than with BDQ susceptibility as suggested by the authors. The cutoff used in the CRyPTIC consortium for BDQ resistance is 0.25 mcg/mL. Looking at figures 1a and 1b, one can see that INH resistant isolates do have a fatter tailed distribution on the right-hand side of these curves. All of this seems to paint a picture that INH resistance is associated with more of a bimodal type effect: for BDQ susceptible isolates, INH resistance is associated with a BDQ MIC that is even slightly further below the BDQ breakpoint. However, INH resistance is associated with a higher risk of a BDQ MIC that is at or above the breakpoint. The clinical significance of all of this is hard to predict. It may well be the case that BDQ treatment is more likely to be successful in patients infected with a strain that has an MIC far below the breakpoint as compared to a strain only slightly below the breakpoint. However, the magnitude of this advantage may well be quantitatively less than the magnitude of the disadvantage found when using BDQ treatment on a strain with an MIC above the breakpoint. And it is clear that there is no major barrier to the emergence of strains which are resistant to both INH and BDQ, even in a set of strains collected before the start of clinical use of BDQ in humans, and thus even in the absence of strong selective pressure for BDQ-resistant strains. The experiment described in lines 82-92 and with data shown in figure 1c may help clarify some of these questions. While there appears to be a trend towards faster death in the INH resistant isolates, statistics are not shown. Given that the authors describe the results by stating "INH-resistant cells trended towards BDQ hyper-susceptibility over INH-susceptible cells", one suspects this did not reach commonly accepted cutoffs for statistical significance. Unfortunately, this question seems to lie very close to the heart of the overall clinical and epidemiological significance of this study. Does the relationship between laboratory-evolved or laboratory-engineered INH-resistance and susceptibility to BDQ truly reflect a phenomenon that is seen in clinical strains and which can have real impact on global TB treatment strategies, or is it an interesting phenomenon without much greater significance? It seems like it would be worth repeating this (or a similar) experiment with a larger random sample of INH-resistant and INH-susceptible strains in order to clarify this question. This is especially relevant given that most of the experiments described in this paper use either a katG E553K mutation or a wholesale deletion of katG (and maybe other neighboring bits of the genome?) which clearly do not accurately recapitulate the dynamics of clinical INH-resistant strains (as seen in the TMP and/or SMP experiments described in lines 264-269 and with data shown in figure 5d).

(3) To the extent it is specified, the dose of BDQ used in these studies is very much on the high side. BDQ in patients is given 400 mg once a day for the first two weeks of therapy and then 200mg qd three times a week in patients. During the first two weeks the Cmax approaches 2 ug/ml for only about four hours per day, during the remainder of the time on treatment (200mg qd) the Cmax is around 0.2ug/ml so this concentration of drug seems high and the differential response would only be relevant for the first two weeks of treatment, (PMID: 38161267). Furthermore, for some experiments each experiment changes the dose of BDQ, for protein carbonylation they use 0.68ug/ml, for 8-oxo-dG they use 5.4ug/ml. How were these doses selected, why not stick with the same dose as the phenotype was observed with?

(4) (line 92) The authors seem to imply that BDQ is only useful for MDR-TB. While BDQ is used in MDR-TB currently there is no clinical data to support that this is due to INH-R/S status. The authors seem to presume that BDQ would not work in INH-S treatment which, given the very slight differences in MIC/Kill is not at all convincing. New regimens for treating fully drug-susceptible TB are in development that include BDQ or other ATP synthase inhibitors.

(5) (Line 89 referring to Fig. 2b (which should be Figure 1C, all of the text figure references in this section are wrong and need to be fixed)). Statistics and reproducibility? It should be noted that H37Rv is barely killed in this assay (<1 log) compared to ca 3 log in similar assays by others (PMID: 29061760, 24569628).

(6) (line 104, referring to Fig. 1c which should be 1d). I don't see any evidence for "synergistic lethality". This effect would be true of ANY companion drug with INH like rifampicin, it is simply suppression of emergence of resistance.

(7) (line 112) The mutant of katG they obtained in vitro was labeled E553K. This is labeled katG- but is it? I can't find where this mutation has ever been observed clinically and certainly no functional analysis of the resulting mutant protein. The overwhelming majority of KatG mutants are S315T as they point out, so why wasn't this mutant (with known catalytic properties) used for these studies?

(8) The entirety of this first section was basically already reported in reference 18 and they cite this as supporting those observation but this does represent some compromise to novelty in the most important part of this manuscript.

(9) (line 153) 7902 BDQ vs 8245 BDQ don't look very different and are the only comparison that has no statistically significant difference between them. Why does 7902 have a higher basal ROS level than 8245? Why the sudden switch to a new strain background? These auxotrophs are different in many ways.

(10) (line 189) In comparing these transcriptional alterations BacTitre glo is a poor substitute for CFU and only suggests intracellular ATP levels are different, NOT that cell viability is affected.

(11) Figure 5 and the section on folate metabolism in many ways highlights the major problem with this manuscript. It is unsurprising that the effect of a complete katG deletion on folate sensitivity doesn't phenocopy to clinical strains because clinical strains rarely have completely disfunction KatG. This study needs to be redone with a single comparator strain – with the katG S315T mutation in an isogenic background to make a plausible case for clinical relevance, and at a consistent dose that is clinically relevant. The constant changing of strains and doses is a sure recipe for misleading and

uninterpretable results.

(12) What is meant by “synergy” is often confusing. For example, in the time kill experiments described in lines 93-99, the authors describe there being “synergistic lethality between BDQ and INH over the 30 day experiment in a dose-dependent manner”. Using the standard CLSI definition of synergy in time kill experiments (which is that the combination has at least 100-fold fewer CFUs than either drug on its own at a given time-point), only the highest tested dose of BDQ can be described as synergistic with INH, and only at the 30 day time-point. Describing synergy as dose-dependent suggests that synergy can be demonstrated at multiple doses, but it stronger at higher doses, which seems not to be the case in the data shown here. On lines 176-179 and 196-198, the authors describe the effects of katG knockout and BDQ on atpE transcript, and describe this as causing “synergistic repression”. I think most readers would interpret this as meaning that while either katG knockout or BDQ treatment alone cause repression in atpE transcript, the combination causes a substantial further degree of repression. However, the data shown indicates that either katG knockout or BDQ treatment alone causes an increase in atpE transcript, but that the combination instead causes a decrease. This is an interesting observation, but I think calling it a “synergistic repression” is likely to lead most readers to misunderstand the phenomenon, especially given that the actual data for this is in the supplemental information.

Reviewer #3

(Remarks to the Author)

The authors of this study have shown that clinical multi-drug resistant strains of *M. tuberculosis* are more likely to be sensitive to Bedaquiline (BDQ) and that catalase deficiency as a result of KatG mutations (conferring isoniazid (INH) resistance) is a contributing factor to this. In addition, they have experimentally demonstrated using KatG mutants that catalase deficiency results in an increased sensitivity to BDQ in lab adapted strains.

They provide evidence of the mechanism by which catalase deficiency could lead to increased sensitivity to BDQ; increased susceptibility to reactive oxygen species, susceptibility to DNA damage and transcriptional and metabolic remodelling.

Collectively their findings are of significance to the mycobacterial drug discovery field because they demonstrate how changes induced by drug resistance mutations can lead to vulnerabilities which could be targeted by other compounds/antibiotics. The study also further re-iterates the importance of studying and understanding the mechanisms of synergistic effects of drug combinations being used clinically.

Sufficient evidence is provided for the conclusions that the authors reach. However, the authors should define the term ‘hyper-susceptible’ because in some cases their results show a moderate or slight increase in sensitivity to BDQ as opposed to hyper-susceptibility.

The quality of the data is good, the appropriate techniques were utilised and enough detail is provided in the methods for the work to be reproduced.

Major comments

- The work is premised on the observation that INH-resistant clinical isolates are hypersensitized to BDQ due to KatG loss of function. From the work, it appears to be the case. But the manuscript would benefit from discussion of potential alternative explanations, where INH resistance is enriched in particular *Mtb* lineages. For example, *inhA* promoter mutations are enriched in lineage 1 whereas KatG-S315T mutations are enriched in lineage 2 (PMCID: PMC3370767); are lineage 1 MDR isolates also hypersensitized to BDQ?
- How were the 5 INH-resistant strains chosen to ensure randomness? (line 82)
- The agar plating experiment is not convincing (lines 104-113 and Fig 1d). The conclusions of the overall work are not altered, but since treatments with BDQ result in low CFU, the INH-resistant CFU are below the limit of detection. BDQ treatment alone might not suppress INH resistance directly, but INH-resistant cells will be killed by BDQ (or any drug other than INH). With INH treatment, INH-resistant cells can expand to fill the capacity left by killed sensitive cells, resulting in a large INH-resistant CFU. Nevertheless, Fig. 1e successfully makes the point that katG loss of function causes sensitivity to BDQ. The authors should consider removing Fig. 1d.
- If Fig. 1d is retained, a no antibiotic control should be included to compare the results in Fig 1D showing the synergistic effects of INH and BDQ. Additionally, 2.7 ug/ml was chosen as the baseline concentration to investigate the mechanisms of BDQ sensitivity across the study, but a reason for this was not stated in the manuscript.
- Likewise, 5.4 ug/ml of BDQ was used to demonstrate reactive oxygen species formation in KatG-deficient mutants (Fig 2/ lines 144-168) but the reasoning behind this is not clearly stated.
- The authors should mention that the E553K mutations (line 112) are probably a result of picking sibling colonies. They should also quantify the level of resistance of their katG-E553K strain, and note its relevance to clinical resistance.
- The full genotype of auxotrophic strains mc28245 and mc27902 should be mentioned in the main text. Would the auxotrophy be expected to change the metabolism and affect the results?
- Analysis and interpretation of dose response curves requires more discussion, especially where they do not strongly support the claims in the text.
 - o For example, the differences observed in the MIC BacTiter-Glo and the OD600 results in Fig 3C& D is not discussed, but the authors conclude (lines 188-190) that overexpression of *kmtR* and *Rv3160c* sensitized the strains to BDQ. In particular, for OD600 measurement of *kmtR*, there appears to be little if any sensitization.
 - o The BacTiter-Glo assay depends on cellular ATP, but is used to measure viability in the presence of the ATP synthase inhibitor BDQ. How would that influence the observed results?
 - o Dose response curves should also be normalized to show percentage growth (or inhibition). For example, Fig. 1g may not show increased susceptibility if normalized.

o In the right-hand panel of Fig. 5b, there is not susceptibility of the mutant to SMX-TMP. The authors should discuss why that might be.

Minor comments

- The results section would flow better if the figures were arranged in a similar order and numbered accurately. For example, in lines 89-172, the figures (Fig.1) cited do not currently correspond to the figures they refer to.
- What is the breakpoint for INH resistance used in this study?
- Methodology should be mentioned in figure legends or the text where they are referenced. For example, it is not clear how gene expression in Fig. 2b, 3a, 3b, 4a etc are measured.
- Fig 3- legend needs to be edited as the captions do not match.
- The scales on Fig 4 should be edited to match each other for clarity and comparison
- The discussion section could also benefit from having the figures mentioned where they are discussed. They are referred to in a few instances.
- The authors correctly note that TMP and SMX are not used in TB treatment, but should note that the antifolate PAS was part of the first combination therapy for TB. It would strength the manuscript to repeat the experiments of Fig. 5 with PAS, but it is not essential.

Reviewer #4

(Remarks to the Author)

Reviewer #5

(Remarks to the Author)

Version 1:

Reviewer comments:

Reviewer #1

(Remarks to the Author)

The authors have addressed my concerns.

Reviewer #2

(Remarks to the Author)

The authors have substantially revised the paper and provided the additional new experiments I requested in the prior review.

Reviewer #3

(Remarks to the Author)

The authors have addresses all comments, and the manuscript has improved dramatically in the clarity of the results and support of the authors' conclusions.

Reviewer #4

(Remarks to the Author)

RESPONSE TO REVIEWER COMMENTS

NCOMMS-24-00092: “Catalase deficiency sensitizes multidrug-resistant *Mycobacterium tuberculosis* to the ATP synthase inhibitor bedaquiline”

We appreciate the reviewer’s thoughtful, helpful, and constructive comments on our manuscript. In response, we have performed several new experiments to address the reviewers’ concerns and to improve the manuscript. These experiments include new complementation studies for BDQ, INH, and H₂O₂ susceptibility with a H37Rv $\Delta katG::pkatG$ strain (Figures 1d, 2a, Supplementary Fig. 1a, and 1b) and H₂O₂, phleomycin, TMX, and SMX susceptibility in a non-MDR INH^R clinical strain containing a *katG* S315T mutation (Figures 2b, 4d, 5b, 5c, and Supplementary Fig. 2a, 4b, and 6c).

These data not only support, but also provide further insights into our claims that the physiological consequences of catalase activity loss-of-function in multidrug-resistant *M. tuberculosis* (Mtb) sensitize Mtb to bedaquiline. We have also thoroughly corrected several textual errors that were present in our original submission. We are confident that these additional data and textual revisions address the reviewers’ concerns and significantly strengthen our manuscript, as described point-by-point below.

Reviewer 1

This manuscript by Ofori-Anyinam investigates the impact of isoniazid resistance in *Mycobacterium tuberculosis* strains on the susceptibility towards bedaquiline. BDQ represents a central component of the recently established BPaLM MDR-TB treatment regimen and therefore gaining insight into factors underlying the sensitivity of INH-resistant strains towards BDQ is of key importance.

The authors initially perform an extensive survey of >10,000 *Mycobacterium tuberculosis* clinical isolates and report that INH-resistant strains, compared to INH-sensitive strains, on average display decreased MIC values for BDQ (similar for MDR-strains versus non-MDR strains). Extending previous studies, the authors find that addition of BDQ suppressed the development of resistance against INH and that various KatG mutant strains display enhanced BDQ sensitivity in time kill kinetics. KatG loss of function seems to be the parameter that causes the enhanced BDQ susceptibility in MDR strains.

KatG deficient strains also showed enhanced sensitivity for H₂O₂, but not for uncouplers. In contrast to previous results, the authors found that BDQ induced ROS formation, which was enhanced in KatG deficient strains. This also translated into cellular damage such as increased protein carbonylation and may contribute to the observed BDQ hypersensitivity.

Based on RNAseq experiments the authors test if differentially regulated components, such as transcription factors and proteins involved in DNA repair can influence BDQ sensitivity. Indeed, over-expression of *kmtR* and *Rv3160* sensitized the bacteria to BDQ and KatG deficient bacteria, including INH-resistant and MDR clinical isolates, were found more susceptible to DNA damage.

Finally, a genome-scale metabolic model for KatG deficient bacteria was applied to predict metabolic flux changes as compared to wild-type bacteria. The model indicated, in contrast to the RNAseq, decreased folic acid metabolism. The authors experimentally support this prediction and demonstrate increased vulnerability of KatG deficient bacteria for SMX and TMP, two

inhibitors of the folic acid biosynthesis pathway (though not for the combination of these two inhibitors).

The picture emerges that KatG deficiency is connected to BDQ hyper-susceptibility by enhanced ROS formation, impaired DNA repair, transcriptional remodeling and folic acid biosynthesis inhibition.

These results are important and will have a significant impact on the field.

We thank the reviewer for their enthusiasm and kind comments.

Please address the following points:

Major points:

1. Figure numbers mentioned in main text in a number of cases do not seem to be accurate, see examples below. This makes navigating the paper difficult.

Line 88-89: Fig 1c meant instead of Fig 2b?

Line 104: Fig 1d instead of Fig 1c?

Line 115: Fig 1e instead of 1 d?

Line 118: 1f instead of 1e?

Line 126: Fig 1i instead of Fig 2h?

Please check throughout the paper.

We have carefully corrected all errors on figure references throughout the manuscript. Some figures have changed and all figure references in the text should now be correct.

2. The authors use a variety of KatG deficient strains (large deletion, single gene deletion, point mutation), making it likely that an observed phenotype is really caused by deletion of KatG. Nevertheless, it is recommended to support this conclusion by a complementation study, expressing KatG gene again in the deficient strain. Including such a complementation in one of the presented experiments seems sufficient.

We thank the reviewer for this recommendation. We have complemented *katG* into our $\Delta katG$ strain and repeated the bedaquiline (BDQ) dose-response experiments in *katG* complement and empty vectors. We validated our complemented strain using INH and H₂O₂ growth inhibition experiments. Our new data further support our core hypothesis that catalase activity deficiency underlies BDQ hyper-susceptibility in drug-resistant Mtb. These new data are presented in Figures 1d, 2a, Supplementary Fig. 1a, and 1b.

3. Lines 104-106 and elsewhere in manuscript: can the authors describe which measures were taken to minimize drug carry over in the time kill experiments? This seems exceptionally important in case of BDQ, where low concentrations carried along to the agar plate may already prevent bacterial growth.

In these experiments, we did not take specific measures to minimize BDQ carry-over during the 30-day time-kill experiments. However, we do not think there is enough BDQ carry-over occurring in our experiments to inhibit growth on 7H10 agar for the following reasons:

1. The reported BDQ MIC for Mtb H37Rv is 0.030 – 0.120 $\mu\text{g}/\text{mL}$ (Andries K, *Science* 2005). The BDQ MIC for the WT H37Rv strain used in this study is 0.240 $\mu\text{g}/\text{mL}$. At the concentration used in our time-kill experiments (2.7 $\mu\text{g}/\text{mL}$), two to three 10-fold dilutions were required to enumerate CFUs which were $\sim 10^5$ - 10^7 CFUs on Day 0. The expected carry-over BDQ concentrations under these conditions would be 0.0027-0.027 $\mu\text{g}/\text{mL}$, which is well below the MIC.
2. We observed up to 4-5 logs of killing in INH^R and MDR strains and 1-2 logs of killing in WT and INH^S strains after 30 days of treatment. Assuming no BDQ instability over this length of time, the expected carry-over BDQ concentrations for each sample would be 2-6 orders of magnitude below the MIC.

Minor points:

Line 50: INH-resistant strains typically carry a mutation in KatG. Can the authors provide information if these mutations only prevent the activation of the INH prodrug or if they completely abolish the catalase activity of this enzyme?

We unfortunately did not directly measure INH activation in these strains. However, four of our drug-resistant Mtb clinical strains possessed KatG S315T mutations, which Reviewer 2 noted does not completely abolish catalase-peroxidase activity. KatG_{S315T} catalase activity is ~ 6 -fold lower than WT KatG and INH activation is ~ 7 -fold decreased (Wengenack, *Journal of Infectious Disease* 1997). We therefore expect both decreased catalase activity and decreased INH activation to be taking place in our INH^R strains. To verify that the INH^R clinical strains used in our study lost catalase activity, we performed new H₂O₂ growth inhibition experiments with all clinical strains and found nearly all of them to be sensitized to H₂O₂. These new data are presented in Figures 2b, and Supplementary Fig. 2a.

Because INH activation is irrelevant for BDQ susceptibility, we think *katG* deletion mutants adequately model the key physiological mechanisms that sensitize INH^R strains to BDQ. In support of this claim, our non-MDR *katG* S315T INH^R clinical strain shared all susceptibility phenotypes with H37Rv $\Delta katG$ (BDQ, H₂O₂, phleomycin, SMX, TMP) (Figures 2b, 4d, 5b, 5c, and Supplementary Fig. 2a, 4b, and 6c).

Line 125: “reduction of CFU over uninduced cells”

This text probably refers to Fig 1i, however, in that panel bacteria carrying the *pfurA* plasmid are compared to bacteria with empty plasmid. Please clarify. In Fig 1i, the reduction in CFU seems stronger than the described modest 1-2 log.

We have corrected and clarified this sentence with the following:

Lines 112-114: “We found *katG* repression by FurA over-expression resulted in 5-log less CFUs than H37Rv cells expressing an empty vector after 16 days treatment with 2.7 μ g/mL BDQ (Fig. 1f).”

Line 126-127: something missing in this sentence.

We have corrected this sentence with the following:

Lines 178-179: “Collectively, these results demonstrate that deficiencies in catalase activity sensitize INH-resistant and MDR Mtb to BDQ.”

Lines 164-165: “greater increases in BDQ-induced protein carbonylation (Fig. 2f) and deoxyguanosine oxidation (Fig. 2g)” In Fig 2g, the increase upon BDQ addition actually seems lower in 8245 cells as compared to 7902 cells. Please clarify.

We think Reviewer 1 is referring to Figure 2f and not Figure 2g, where the increase in BDQ-induced protein carbonylation appeared less in mc²8245 cells than in mc²7902 cells. We think there are multiple reasons why this may be the case:

1. The level of protein carbonylation was already higher in untreated mc²8245 cells than in mc²7902 cells, so it is possible that this elevated baseline may already be asymptotically close to the biological limit.
2. It is not clear that rates of protein carbonylation and deoxyguanosine oxidation should be proportional, since the expression of proteases and DNA repair enzymes that can remove carbonylated proteins and oxidized deoxyguanosine are not likely to be proportional under BDQ-treatment.

3. It is possible that protein carbonylation in these experiments may have occurred in a non-linear measurement region of these ELISA experiments.

We did not perform new experiments to specifically differentiate between these potential explanations, but we think our collective data from Figure 2e, 2f, and 2g support our hypothesis that BDQ treatment increases total ROS and oxidative cellular damage in INH^R and MDR cells deficient in catalase activity in relation to INH^S cells.

We have revised the manuscript with the following text to specifically clarify that our data indicate greatest total protein carbonylation rather than greater increases in protein carbonylation:

Lines 162-166: “Consistent with the CellROX experiments, protein carbonylation (Fig. 2f) and deoxyguanosine oxidation (Fig. 2g) were greater in mc²8245 cells than in mc²7902 cells following treatment with BDQ. Importantly, protein carbonylation and deoxyguanosine oxidation was also greater in untreated mc²8245 cells than mc²7902 cells, supporting the expectation that catalase-deficient mc²8245 cells would have greater basal oxidative cellular damage than catalase-replete mc²7902 cells.”

Line 173-175: Please confirm that the increased expression mentioned in this sentence was measured in the KatG deficient bacteria.

We have revised the text as follows:

Lines 174-176: “Interestingly, expression of ATP synthase genes was greater and expression of mycolic acid biosynthesis genes was lower in $\Delta katG$ cells than in wild-type cells.”

Line 174-175: “BDQ treatment synergized with katG deletion to further suppress inhA expression.” How was synergy defined or calculated here?

As Reviewer 2 noted, we may not have appropriately used the term “synergy” in the text. To avoid confusion, we have revised the text throughout to minimize use of this term, as follows:

Lines 174-180: “Interestingly, expression of ATP synthase genes was greater and expression of mycolic acid biosynthesis genes was lower in $\Delta katG$ cells than in wild-type cells. Moreover, BDQ further suppressed *inhA* expression in $\Delta katG$ cells than in wild-type cells (Supplementary Fig. 3a). We expected BDQ treatment to further amplify expression of *atpE* (ATP synthase subunit c, the target of BDQ) in $\Delta katG$ cells than in wild-type cells, but instead found that BDQ treatment reduced *atpE* expression in $\Delta katG$ cells to levels similar to untreated wild-type cells (Supplementary Fig. 3b).”

Lines 180-182: “Because INH inhibition of InhA and BDQ inhibition of ATP synthase are both bactericidal for Mtb, these results suggest that BDQ-induced repression of *inhA* and *atpE* expression may also sensitize cells deficient in catalase activity to BDQ.”

Lines 253-255: “Consistent with enhanced BDQ-induced repression of *inhA* expression in $\Delta katG$ cells relative to wild-type cells (Supplementary Fig. 3a), model simulations predicted decreased mycolic acid biosynthesis (Supplementary Fig. 5b).”

Lines 269-272: “Consistent with these results, model simulations predicted that BDQ treatment further suppresses nucleotide metabolism in $\Delta katG$ cells relative to wild-type cells, including downregulation of PRPP synthase and purine and pyrimidine biosynthesis reactions (Supplementary Fig. 6b).”

Lines 202-203: please mention that BDQ was earlier shown to downregulate these pathways (Koul et al 2014, cited elsewhere as ref 11).

We thank Reviewer 1 for this suggestion. We have revised the text with the following:

Lines 215-216: “These results are consistent with previous studies in which BDQ-induced changes in H37Rv gene expression was profiled using microarrays¹¹.”

Lines 264-265 and Fig 5b: any thoughts why the SMX/TMP combination does not act stronger against the KatG deficient strain as compared to wild-type (while the individual inhibitors do act stronger)?

We think the reason why $\Delta katG$ does not have increased susceptibility to SMX-TMP over WT cells is because WT cells are already very sensitized to TMP-SMX when compared with SMX or TMP sensitivity alone. We think TMP-SMX sensitivity has already saturated in these experiments, such that catalase deficiency cannot enable further sensitization. We have added this explanation to our revised manuscript as follows:

Lines 287-290: “Moreover, while $\Delta katG$ and wild-type cells were both sensitized to the combination of TMP and SMX (Supplementary Fig. 6d) relative to TMP alone (Fig. 5b), $\Delta katG$ cells did not exhibit any further TMP-SMX sensitization than wild-type cells, possibly due to saturation of the growth inhibition that can be achieved by folate biosynthesis inhibition.”

Remarks on code availability: This is not my field.

Code for the RNA sequencing analyses and genome-scale metabolic modeling is now deposited in the following GitHub repository: https://github.com/jasonhyang/OforiAnyinam_2024 . Raw sequencing data has now been deposited in the following Sequence Read Archive BioProject: PRJNA1139169.

Reviewer 2

Review of NCOMMS-24-00092: Ofori-Anyinam et al., “Catalase deficiency sensitizes multidrug-resistant Mycobacterium tuberculosis to the ATP synthase inhibitor bedaquiline.”

This manuscript describes studies to understand the mechanistic link between isoniazid resistance and bedaquiline susceptibility in Mtb. The authors begin with a re-analysis of the MIC data collected by the CRyPTIC consortium and identify a slight skew in MIC towards more sensitive for bedaquiline in INH-R Mtb compared to INH-S Mtb. They then recapitulate a slight difference in the BDQ killing kinetics in INH-S vs INH-R strains in another set of a few isolates over 30 days. They spend most of the paper describing studies of laboratory-generated katG mutants to tie this effect to oxidative stress, transcription factors, DNA damage and folate

metabolism. On balance there may be a slight effect but I remained unconvinced this is clinically meaningful or experimentally proven.

We thank the reviewer for their thoughtful concerns and critiques. We have performed several new experiments using a non-MDR INH^R *katG* S315T clinical strain which further support our original findings (Figures 2b, 4d, 5b, 5c, and Supplementary Fig. 2a, 4b, and 6c). We think these new experiments, as suggested by Reviewer 2, significantly strengthen our manuscript.

Specific Comments:

(1) Clinically the most abundant *katG* mutant observed is the S315T mutant (estimates range from 70-90% of INH-R isolates are this mutation) which is far from catalase deficient (see PMID: 9291321 among many others). This mutation retains virtually all the catalase function while decreasing the affinity for INH and imparts very little fitness cost to the organism. Although the authors mention this mutant in Figure 1C in the context of the clinical strains, they go on to study laboratory generated *katG* null mutants which is not at all similar to what is seen clinically. Thus, the relevance of the work to the use of BDQ in MDR-TB patients seems a bit of a stretch.

Reviewer 2 notes that a *katG* S315T mutation does not completely abolish catalase-peroxidase activity. However, the paper that Reviewer 2 refers to (PMID #9291321: Wengenack, *Journal of Infectious Disease* 1997) state that KatG_{S315T} catalase activity is decreased ~6-fold when compared with WT KatG (third sentence of abstract). This is a ~85% decrease, which is far from WT activity. Although it is true that a complete $\Delta katG$ deletion is not the same as a KatG S315T mutation, we think *katG* deletion is a useful genetic model for studying INH resistance.

To address Reviewer 2's concerns on the clinical relevance of our findings, we have repeated many of the key experiments in our study using a non-MDR INH^R *katG* S315T clinical strain (Figures 2b, 4d, 5b, 5c, and Supplementary Fig. 2a, 4b, and 6c). Importantly, our new data show that INH^R clinical strains are sensitized to H₂O₂, indicating they are deficient in catalase activity, even if the loss of catalase activity is not of the same magnitude as total *katG* deletion. These new data are presented in Figures 2b, and Supplementary Fig. 2a. Our collective new data with the non-MDR INH^R *katG* S315T clinical strain support our findings from the $\Delta katG$ mutant strain and also provide new insights on differences between MDR and non-MDR drug-susceptibility phenotypes (Figures 5b, 5c, and Supplementary Fig. 6c).

(2) The prevalence of BDQ and INH resistant isolates in the CRyPTIC consortium data set is not straightforward to interpret. As the authors state (lines 93-94), among INH resistant isolates in this dataset, BDQ resistance is less common than is resistance to any of the other antibiotics

tested – 1.5% of INH resistant isolates were also BDQ resistant (taken from Table G in the supplementary information of reference 13). However, this may be more a reflection of the fact that BDQ has the lowest rate of resistance of any antibiotic in this data set (at 0.9%, taken from page 3 of reference 13), rather than a specific relationship between INH and BDQ resistance. A cursory analysis of the numbers reported in reference 13 (including INH resistance rate of 49%, also taken from page 3 of reference 13) suggests that while 1.5% of INH resistant isolates are also resistant to BDQ, only 0.3% of INH susceptible strains are resistant to BDQ. Thus it actually appears that INH resistance is more associated with BDQ resistance, rather than with BDQ susceptibility as suggested by the authors. The cutoff used in the CRyPTIC consortium for BDQ resistance is 0.25 mcg/mL. Looking at figures 1a and 1b, one can see that INH resistant isolates do have a fatter tailed distribution on the right-hand side of these curves. All of this seems to paint a picture that INH resistance is associated with more of a bimodal type effect: for BDQ susceptible isolates, INH resistance is associated with a BDQ MIC that is even slightly further below the BDQ breakpoint. However, INH resistance is associated with a higher risk of a BDQ MIC that is at or above the breakpoint. The clinical significance of all of this is hard to predict. It may well be the case that BDQ treatment is more likely to be successful in patients infected with a strain that has an MIC far below the breakpoint as compared to a strain only slightly below the breakpoint. However, the magnitude of this advantage may well be quantitatively less than the magnitude of the disadvantage found when using BDQ treatment on a strain with an MIC above the breakpoint. And it is clear that there is no major barrier to the emergence of strains which are resistant to both INH and BDQ, even in a set of strains collected before the start of clinical use of BDQ in humans, and thus even in the absence of strong selective pressure for BDQ-resistant strains. The experiment described in lines 82-92 and with data shown in figure 1c may help clarify some of these questions. While there appears to be a trend towards faster death in the INH resistant isolates, statistics are not shown. Given that the authors describe the results by stating “INH-resistant cells trended towards BDQ hyper-susceptibility over INH-susceptible cells”, one suspects this did not reach commonly accepted cutoffs for statistical significance

Although we understand Reviewer 2’s concern on the association between INH and BDQ resistance, we disagree with their interpretation for the following reasons:

1. As Reviewer 2 notes, and as articulated in the original publication, 99.1% of the strains characterized by the CRyPTIC Consortium were BDQ-susceptible. It is therefore difficult to make strong claims on BDQ resistance from these data as only 0.9% of the total dataset exhibited BDQ-resistance. While we do not think it is misleading to state that the CRyPTIC Consortium’s data show that BDQ resistance is least common in INH^R clinical strains than strains resistant to any of the other 11 drugs tested, we have removed the sentence Reviewer 2 noted (previous Lines 93-94) to eliminate confusion.
2. Although the distributions in BDQ MIC are highly overlapping between INH^R and INH^S strains in Figures 1a and 1b, non-parametric Mann-Whitney statistical testing show that INH^R BDQ MICs are lower than INH^S BDQ MICs with a p-value of $8.95 \cdot 10^{-45}$. These statistical analyses have been added to the text as follows:

Lines 78-81: “We found a global reduction in BDQ MICs at the population level for MDR isolates ($n = 3,958$) in relation to non-MDR isolates ($n = 7,761$) ($p = 3.13 \cdot 10^{-50}$ by Mann-

- Whitney) (Fig. 1a). Similarly, BDQ MICs were decreased for INH-resistant isolates (n = 5,078) in relation to INH-susceptible isolates (n = 5,986) ($p = 8.95 \cdot 10^{-45}$ by Mann-Whitney) (Fig. 1b).”
- Our time-kill data in Figure 1c demonstrate that all strains tested exhibited clear 30-day BDQ susceptibility (greater than 1-log killing) except for one INH^S clinical strain. This is consistent with the CRyPTIC MIC data in which 99.1% of all strains were BDQ-susceptible. We appreciate Reviewer 2’s suggestion that the data in Figure 1c may help clarify their questions and their concern that differences in BDQ susceptibility were not statistically significant between INH^R and INH^S strains. We performed new statistical analyses on the 30-day survival fractions for the data shown in Figure 1c. These analyses show that the percent survival as determined from CFUs before and after BDQ treatment was significantly lower in INH^R than INH^S strains with a p-value of 0.0317, as determined by non-parametric Mann-Whitney test. Note that the black INH^S point is WT H37Rv and the red INH^R point is the non-MDR *katG* S315T INH^R clinical strain. These new results are now presented in Figure 1c.

- While we agree with Reviewer 2 that the decreased BDQ MIC in INH^R clinical strains in relation to INH^S clinical strains in the CRyPTIC dataset is modest, we do not think this modest decrease in BDQ MIC is clinically insignificant. Colangeli et al, *New England Journal of Medicine* 2018 (PMID #30157391) show that modestly increased INH and RIF MICs below the resistance breakpoint was sufficient for predicting relapse TB infection with a ROC AUC of 0.875. INH and RIF MICs for all strains included in that NEJM study were below the INH and RIF resistance breakpoints of 0.8 and 1.0 $\mu\text{g/mL}$ (as defined by the CRyPTIC Consortium), respectively. Although, as Reviewer 2 notes, it remains to be seen if the modest increases in BDQ MICs in INH^S over INH^R clinical strains will cause relapse infection, we think the data by Colangeli et al. support the potential clinical relevance of our study. We have added new text to address this point in the discussion as follows:

Lines 354-362: “Third, our analyses of CRyPTIC drug susceptibility data revealed only modest decreases in BDQ MIC for INH-resistant clinical strains relative to INH-susceptible strains at the population level (Fig. 1b). Although the clinical impact of small differences in BDQ MICs between INH-susceptible and drug-resistant clinical strains is not yet clear, our results demonstrate that BDQ lethality in catalase-deficient cells can be significant (Fig. 1e) despite small changes in MIC (Fig. 1d). Relatedly, recent findings demonstrate that sub-breakpoint differences in INH and RIF MIC are sufficient for predicting TB reinfection after 6-months first-line chemotherapy⁴⁷. These results highlight how important insights into drug-susceptibility and/or drug-resistance may be overlooked by conventional approaches only measuring MICs”

Unfortunately, this question seems to lie very close to the heart of the overall clinical and epidemiological significance of this study. Does the relationship between laboratory-evolved or laboratory-engineered INH-resistance and susceptibility to BDQ truly reflect a phenomenon that is seen in clinical strains and which can have real impact on global TB treatment strategies, or is it an interesting phenomenon without much greater significance? It seems like it would be worth repeating this (or a similar) experiment with a larger random sample of INH-resistant and INH-susceptible strains in order to clarify this question. This is especially relevant given that most of the experiments described in this paper use either a *katG* E553K mutation or a wholesale deletion of *katG* (and maybe other neighboring bits of the genome?) which clearly do not accurately recapitulate the dynamics of clinical INH-resistant strains (as seen in the TMP and/or SMP experiments described in lines 264-269 and with data shown in figure 5d).

As stated above, we have repeated key experiments in our study using a non-MDR INH^R *katG* S315T clinical strain (Figures 2b, 4d, 5b, 5c, and Supplementary Fig. 2a, 4b, and 6c). These data strongly support our results from the H37Rv $\Delta katG$ strain.

(3) To the extent it is specified, the dose of BDQ used in these studies is very much on the high side. BDQ in patients is given 400 mg once a day for the first two weeks of therapy and then 200mg qd three times a week in patients. During the first two weeks the C_{max} approaches 2 ug/ml for only about four hours per day, during the remainder of the time on treatment (200mg qd) the C_{max} is around 0.2ug/ml so this concentration of drug seems high and the differential response would only be relevant for the first two weeks of treatment, (PMID: 38161267). Furthermore, for some experiments each experiment changes the dose of BDQ, for protein carbonylation they use 0.68ug/ml, for 8-oxo-dG they use 5.4ug/ml. How were these doses selected, why not stick with the same dose as the phenotype was observed with?

We initially chose 2.7 $\mu\text{g}/\text{mL}$ BDQ because this concentration conferred the strongest killing phenotype in our time-kill experiments combining INH and BDQ (previous Supplementary Fig. 1; below). From these time-kill CFU data and the CellROX ROS measurements in Figure 2e, we retrospectively think we would have found very similar results if we used 0.68 $\mu\text{g}/\text{mL}$ BDQ throughout this project.

Following concerns raised by Reviewers 2 and 3 on the BDQ concentrations used in our experiments, we carefully reviewed laboratory notebooks for all the experiments performed for this project. We found that the 5.4 $\mu\text{g}/\text{mL}$ concentration stated for the ELISA experiments was incorrectly stated in the manuscript. The reason for this mistake was because in the technical execution of the ELISA experiments, 5 mL of mid-log Mtb culture was added to 5 mL 5.4 $\mu\text{g}/\text{mL}$ BDQ to achieve 10 mL Mtb cultures with 2.7 $\mu\text{g}/\text{mL}$ BDQ working concentrations. 0.68 $\mu\text{g}/\text{mL}$ BDQ was used in the CellROX ROS

experiments because higher concentrations of BDQ resulted in measurements beyond the linear range of our calibration curves. We carefully corrected typographical errors for the concentrations used in our ELISA experiments in the figure legends for Figure 2e, 2f and 2g and in the methods sections.

(4) (line 92) The authors seem to imply that BDQ is only useful for MDR-TB. While BDQ is used in MDR-TB currently there is no clinical data to support that this is due to INH-R/S status. The authors seem to presume that BDQ would not work in INH-S treatment which, given the very slight differences in MIC/Kill is not at all convincing. New regimens for treating fully drug-susceptible TB are in development that include BDQ or other ATP synthase inhibitors.

We do not mean to convey that BDQ is only useful for MDR-TB. Indeed, as Figure 1c shows, MDR and non-MDR INH^R and INH^S clinical strains all exhibited BDQ lethality in our 30-day time kill experiments. As stated above, although differences in MIC are small between INH^R and INH^S clinical strains in the CRyPTIC dataset (Figure 1b), the median increase in BDQ killing was at least 2-log in our INH^R vs INH^S clinical strains (Figure 1c). We do not think 2-log increase is insignificant. We agree that inhibitors of ATP synthase and other respiratory chain complexes are an exciting area of active anti-TB drug discovery and we think our results here will be relevant for those efforts.

(5) (Line 89 referring to Fig. 2b (which should be Figure 1C, all of the text figure references in this section are wrong and need to be fixed)). Statistics and reproducibility? It should be noted that H37Rv is barely killed in this assay (<1 log) compared to ca 3 log in similar assays by others (PMID: 29061760, 24569628).

We have carefully corrected all errors on figure references throughout the manuscript. Some figures have changed and all figure references in the text should now be correct.

As stated above, we have now performed statistics on the 30-day time kill experiments depicted in Figure 1c and found that these results are indeed significant. These results are now presented in Figure 1c.

We acknowledge that we only observed ~0.7-log killing in our H37Rv experiments and that these appear to be less than Koul, *Nature Communications* 2014 (PMID 29061760) and Berube, *Antimicrobial Agents and Chemotherapy* 2018 (PMID 24569628). However, we think there are multiple reasons why this may be taking place. Koul et al performed their experiments in unsupplemented 7H9 media while our experiments took place in 7H9 + OADC media. Because we show that ROS is an important source of BDQ susceptibility (Figure 2e), it is possible that the presence of catalase in our media (OADC) may abrogate some of the BDQ lethality in our experiments relative to the prior work. Work by Berube et al support our data as they performed their experiments in Mtb H37Rv under experimental conditions similar to ours and found that H37Rv only exhibited ~1.5-log killing at 21 days (Berube Figure 5B, depicted below). We think their data is similar to our results in Figure 1c.

(6) (line 104, referring to Fig. 1c which should be 1d). I don't see any evidence for "synergistic lethality". This effect would be true of ANY companion drug with INH like rifampicin, it is simply suppression of emergence of resistance.

As noted in response to Reviewer 2's 12th comment below, we think our use of the term synergistic was imprecise. We agree with Reviewer 2's interpretation that these experiments indicate suppression of INH resistance emergence. However, because Reviewer 2 indicated our evolved KatG E553K mutant strain may not be clinically relevant; because Reviewer 3 raised reasonable concerns on the design and interpretation of this experiment; and because this experiment is not central to this work, we have removed these results from our revised manuscript.

(7) (line 112) The mutant of *katG* they obtained in vitro was labeled E553K. This is labeled *katG*- but is it? I can't find where this mutation has ever been observed clinically and certainly no functional analysis of the resulting mutant protein. The overwhelming majority of *KatG* mutants are S315T as they point out, so why wasn't this mutant (with known catalytic properties) used for these studies?

We agree with Reviewer 2 that our evolved *KatG* E553K mutant strain may not be clinically relevant. We have therefore removed this strain from this revision.

(8) The entirety of this first section was basically already reported in reference 18 and they cite this as supporting those observation but this does represent some compromise to novelty in the most important part of this manuscript.

We do not agree with Reviewer 2 that the entirety of our first section was basically already reported in reference 18 (Waller, *Nature Communications* 2023). While Waller et al showed via laboratory evolution in BSL2 *Mtb* auxotroph strains that *katG* mutations sensitize *Mtb* to BDQ, their work (1) did not connect these phenotypes to any evidence in clinical strains – Figures 2b, 4d, 5b, 5c, and Supplementary Fig. 2a, 4b, and 6c; (2) did not give any mechanistic explanation for these phenotypes – Figures 2-5; (3) did not include any *katG* S315T strains – noted extensively by Reviewer 2 to be important for clinical relevance; and (4) was fully based on work in auxotroph strains – noted by Reviewers 2 and 3 to decrease their clinical relevance.

Because our work here mechanistically investigates how catalase activity deficiency enables BDQ sensitization and because we have performed experiments in Mtb clinical strains (including three MDR *katG* S315T strains and a non-MDR *katG* S315T strain), we think our work is both novel and impactful.

(9) (line 153) 7902 BDQ vs 8245 BDQ don't look very different and are the only comparison that has no statistically significant difference between them. Why does 7902 have a higher basal ROS level than 8245? Why the sudden switch to a new strain background? These auxotrophs are different in many ways.

The increase in BDQ-induced CellROX expression is higher in mc²8245 cells than mc²7902 cells which indicates that they are indeed different. We are unsure what Reviewer 2 refers to as “no statistically significant difference” in or around line 153. All statistical comparisons made are depicted explicitly in Figures 2e, 2f, and 2g. The only non-significant differences measured were between untreated and BDQ-treated protein carbonylation levels in mc²8245 and between untreated and BDQ-treated 8-oxo-dG levels in mc²7902. We think these are due to the higher variance in BDQ-treated cells and multiple comparisons false discovery corrections (which is why we did not statistically test all pairwise combinations of measurements). Non-parametric rank-based statistical testing would demonstrate clear statistical significance in these comparisons, but we reasoned that a parametric Welch's t-test (which does not assume equal variances) is a more appropriate, and therefore more rigorous, statistical test to perform.

We do not have a good explanation for why CellROX fluorescence is (modestly) lower in untreated mc²8245 cells than in untreated mc²7902 cells. A plausible reason is that our RNA sequencing data reveal that untreated $\Delta katG$ cells express more ROS detoxifying *ahpC*, *ahpD*, *trxC*, and ergothioneine biosynthesis genes than WT cells (Supplementary Table 3, Lines 173-174) and that these can compensate for the lack of *katG* in unstressed growth conditions. We think that in the presence of a ROS-generating stress such as BDQ, KatG-deficient cells would have a harder time limiting ROS accumulation than KatG-replete cells despite having compensatory expression of other ROS detoxifying genes.

We switched to these auxotroph strains because it was more feasible for our lab to execute the CellROX and ELISA experiments in a BSL2 environment than in a BSL3 environment. While we agree that auxotroph strains are different than WT strains, our experiments in Figures 2d, 4c, 5b, 5c, and Supplementary Fig. 2b show that these auxotrophs strains show similar phenotypes to both WT vs $\Delta katG$ H37Rv strains and INH^S vs INH^R clinical strains. We therefore think these auxotroph strains are a relevant model for this study.

(10) (line 189) In comparing these transcriptional alterations BacTitre glo is a poor substitute for CFU and only suggests intracellular ATP levels are different, NOT that cell viability is affected.

We agree with Reviewer 2 that BacTitre-Glo is not a substitute for enumerating CFUs. However, we wish to clarify that the experiments in Figures 3b, 3c, and Supplementary Fig. 3c were not BDQ lethality experiments and therefore not measurements of (loss of viability). These experiments were instead growth experiments from a low starting cell density. We think that BacTitre-Glo and turbidity (OD₆₀₀) report on two different aspects of Mtb physiology: ATP production vs growth. In the case of *Rv3160c*, the BDQ sensitivity of both ATP production and growth are higher in *Rv3160c* overexpressing cells than in control cells (Figure 3b). In the case of *kmtR*, only ATP production is sensitized to BDQ in *kmtR* over-expressing

cells (Figure 3c). In the case of *prpR*, there is no significant difference between *prpR* and control cells (Supplementary Fig. 3c). We have clarified these in the text as follows:

Lines 191-200: “To test the hypothesis that induction of these programs would sensitize Mtb to BDQ, we performed BDQ growth inhibition dose-response experiments on H37Rv cells over-expressing the three transcription factors with the greatest increases in basal expression in $\Delta katG$ cells relative to wild-type cells (*Rv3160c*, *kmtR*, *prpR*)^{16,17} relative to their empty vector control (Fig. 3a). We measured the BDQ sensitivity for inhibition of ATP synthesis by BacTiter-Glo and BDQ sensitivity for growth inhibition by turbidity (OD₆₀₀). In support of our hypothesis, BDQ-mediated inhibition of ATP synthesis in cells over-expressing *Rv3160c* or *kmtR* was sensitized to BDQ (Fig. 3b and 3c). In addition, *Rv3160c* over-expression sensitized cells to BDQ-mediated growth inhibition. However, we did not observe changes in BDQ sensitivity for either ATP production or growth in *prpR* over-expressing cells (Supplementary Fig. 3c).”

(11) Figure 5 and the section on folate metabolism in many ways highlights the major problem with this manuscript. It is unsurprising that the effect of a complete *katG* deletion on folate sensitivity doesn't phenocopy to clinical strains because clinical strains rarely have completely dysfunction *KatG*. This study needs to be redone with a single comparator strain – with the *katG* S315T mutation in an isogenic background to make a plausible case for clinical relevance, and at a consistent dose that is clinically relevant. The constant changing of strains and doses is a sure recipe for misleading and uninterpretable results.

We performed new analyses to better understand why we did not see consistent antifolate sensitization in INH^R clinical strains relative to INH^S clinical strains. We discovered that in our original experiments, the non-MDR *katG* S315T INH^R clinical strain was sensitized to TMP while the MDR clinical strains were not. We performed new experiments in this *katG* S315T INH^R clinical strain to validate our original findings and found that these results are robust (Fig. 5b, 5c, and Supplementary Fig. 6c; below). We think these results further validate our findings that catalase-deficient cells are sensitized to inhibition of folate biosynthesis. We think these data also suggest that *rpoB* mutations alter Mtb physiology in a way that protects against folate synthesis inhibition and indicate that future studies will need to further probe changes in *Mtb* physiology induced by MDR. We have updated the manuscript to include these additional data and insights as follows:

Lines 279-287: “Consistent with our hypothesis, $\Delta katG$, mc²8245 cells, and a non-MDR INH-resistant clinical strain were all hypersensitive to TMP relative to wild-type, mc²7902, and INH-susceptible clinical strains, respectively (Fig. 5b and Supplementary Fig. 6c). Similarly, each of these cells were also sensitized to SMX relative to their respective controls (Fig. 5c and Supplementary Fig. 6c). Interestingly, although the non-MDR *katG* S315T INH-resistant clinical strain (TDR-TB 42) exhibited increased TMP and SMX sensitivity relative to INH-susceptible strains, MDR clinical strains did not exhibit changes in TMP or SMX sensitivity. These suggest that the mutations in *rpoB* necessary for conferring rifampicin resistance (and thus making INH-resistant strains MDR) compensate for BDQ-induced defects in folate biosynthesis in catalase-deficient cells (Fig. 5a).”

Lines 363-371: “In addition, although our non-MDR INH-resistant strain containing a *katG* S315T mutation (TDR-TB. 42) was sensitized to TMP and SMX relative to INH-susceptible clinical strains, we did not find TMP or SMX sensitization in MDR strains containing *katG* mutations (Fig. 5d and

Supplementary Fig. 6c). It is possible that the differences between MDR and non-MDR INH-resistant clinical strains are due to the additional changes in Mtb physiology caused by the *rpoB* mutations that confer RIF resistance. It is also possible that other genomic mutations naturally enriched in INH-resistant clinical strains (such as over-expression of *ahpC*⁴⁸) may suppress antifolate hypersensitivity by compensating for deficient catalase activity. Our results here underscore the need for more mechanistic work to better understand collateral sensitivity mechanisms in MDR-TB.”

We wish to clarify that in this study we did not constantly change strains and doses. WT and $\Delta katG$ H37Rv cells were used in almost all experiments except for the BSL-2 ROS and oxidative damage experiments depicted in Figure 2. These experiments were performed using auxotrophic *mc*²⁷⁹⁰² and *mc*²⁸²⁴⁵ cells, which we show in Figures 2d, 4c, 5b, 5c, and Supplementary Fig. 2b share the same phenotypes as $\Delta katG$ and non-MDR *katG* S315T INH-resistant clinical cells. We used a panel of INH^S and INH^R clinical strains in Figures 2b, 4d, 5b, 5c, and Supplementary Fig. 2a, 4b, and 6c to validate and demonstrate the clinical relevance of our findings from $\Delta katG$ and *mc*²⁸²⁴⁵ cells.

In addition, as stated above, 2.7 μ g/mL BDQ was used for almost all experiments, except for the CellROX ROS measurements where 0.68 μ g/mL BDQ was used to ensure measurements were within the dynamic range of the assay within our experimental execution. The 5.4 μ g/mL BDQ in some parts of the text were typographical mistakes due to miscommunication. All experiments not involving BDQ were performed at multiple concentrations in consistently shared dose-response experiments. We therefore think our use of strains and concentrations has been consistent throughout the study and that our overall data are largely self-consistent, providing strong evidence for our conclusions.

(12) What is meant by “synergy” is often confusing. For example, in the time kill experiments described in lines 93-99, the authors describe there being “synergistic lethality between BDQ and INH over the 30 day experiment in a dose-dependent manner”. Using the standard CLSI definition of synergy in time kill experiments (which is that the combination has at least 100-fold fewer CFUs than either drug on its own at a given time-point), only the highest tested dose of BDQ can be

described as synergistic with INH, and only at the 30 day time-point. Describing synergy as dose-dependent suggests that synergy can be demonstrated at multiple doses, but it stronger at higher doses, which seems not to be the case in the data shown here. On lines 176-179 and 196-198, the authors describe the effects of katG knockout and BDQ on atpE transcript, and describe this as causing “synergistic repression”. I think most readers would interpret this as meaning that while either katG knockout or BDQ treatment alone cause repression in atpE transcript, the combination causes a substantial further degree of repression. However, the data shown indicates that either katG knockout or BDQ treatment alone causes an increase in atpE transcript, but that the combination instead causes a decrease. This is an interesting observation, but I think calling it a “synergistic repression” is likely to lead most readers to misunderstand the phenomenon, especially given that the actual data for this is in the supplemental information.

We thank Reviewer 2 for detailing our confusing usage of the term “synergy”. We had imprecisely used this term to convey epistasis without rigorously considering the precise definition of this term. We have revised the text throughout to remove our potentially confusing use of the term “synergy”.

Reviewer 3

The authors of this study have shown that clinical multi-drug resistant strains of *M. tuberculosis* are more likely to be sensitive to Bedaquiline (BDQ) and that catalase deficiency as a result of KatG mutations (conferring isoniazid (INH) resistance) is a contributing factor to this. In addition, they have experimentally demonstrated using KatG mutants that catalase deficiency results in an increased sensitivity to BDQ in lab adapted strains.

They provide evidence of the mechanism by which catalase deficiency could lead to increased sensitivity to BDQ; increased susceptibility to reactive oxygen species, susceptibility to DNA damage and transcriptional and metabolic remodelling.

Collectively their findings are of significance to the mycobacterial drug discovery field because they demonstrate how changes induced by drug resistance mutations can lead to vulnerabilities which could be targeted by other compounds/antibiotics. The study also further re-iterates the importance of studying and understanding the mechanisms of synergistic effects of drug combinations being used clinically.

Sufficient evidence is provided for the conclusions that the authors reach. However, the authors should define the term ‘hyper-susceptible’ because in some cases their results show a moderate or slight increase in sensitivity to BDQ as opposed to hyper-susceptibility.

The quality of the data is good, the appropriate techniques were utilised and enough detail is provided in the methods for the work to be reproduced.

We thank the reviewer for their enthusiasm and kind comments.

We have carefully revised our usage of the terms “susceptibility” and “sensitivity” throughout our manuscript. We now exclusively use susceptibility to refer to shifts in lethality from time-kill experiments and now exclusively use sensitivity to refer to shifts in drug concentrations from dose-response experiments.

Major comments

The work is premised on the observation that INH-resistant clinical isolates are hypersensitized to BDQ due to KatG loss of function. From the work, it appears to be the case. But the manuscript would benefit from discussion of potential alternative explanations, where INH resistance is enriched in particular Mtb lineages. For example, *inhA* promoter mutations are enriched in lineage 1 whereas KatG-S315T mutations are enriched in lineage 2 (PMCID: PMC3370767); are lineage 1 MDR isolates also hypersensitized to BDQ?

This is a very interesting point and we have added a discussion on lineage differences to the text as follows:

Lines 349-353: “Moreover, INH-resistance conferring mutations are differentially enriched in different Mtb lineages and it is possible that lineage-specific differences in Mtb physiology may also mechanistically contribute to BDQ susceptibility. For example, *katG* S315T mutations are more prevalent in Lineage 4 strains while *inhA* promoter mutations are more prevalent in Lineage 1 strains⁴⁶. It will be interesting for future studies to uncover how such differences may more broadly impact Mtb susceptibility to other drugs.”

How were the 5 INH-resistant strains chosen to ensure randomness? (line 82)

We randomly selected the INH^R and INH^S clinical strains in this study without examining information on their genotypes, susceptibilities to other antibiotics, lineages, or geographical origins. One of these strains luckily possessed a KatG S315T with no RpoB mutation that we used for in each of

We have clarified this detail in the text as follows:

Lines 85-87: “These strains were randomly selected without first examining information on their genotypes, susceptibilities to antibiotics other than INH, Mtb lineages, or geographical origins (Supplementary Tables 1 and 2).”

The agar plating experiment is not convincing (lines 104-113 and Fig 1d). The conclusions of the overall work are not altered, but since treatments with BDQ result in low CFU, the INH-resistant CFU are below the limit of detection. BDQ treatment alone might not suppress INH resistance directly, but INH-resistant cells will be killed by BDQ (or any drug other than INH). With INH treatment, INH-resistant cells can expand to fill the capacity left by killed sensitive cells, resulting in a large INH-resistant CFU. Nevertheless, Fig. 1e successfully makes the point that *katG* loss of function causes sensitivity to BDQ. The authors should consider removing Fig. 1d.

Although our evolved KatG E553K mutant strain exhibited BDQ hyper-susceptibility in time-kill experiments, we agree with Reviewer 3 that our BDQ MIC experiments between WT and KatG_{E553K} do not appear convincing when normalized. We do think our agar plating experiments support our claim that BDQ co-treatment suppresses the emergence/fixation of INH resistance. However, Reviewer 2 also noted that our evolved KatG E553K mutant strain may not be clinically relevant. In light of these weaknesses, and the confusion over the agar plating experiment, we have decided to remove this laboratory evolution experiment from our revised manuscript.

If Fig. 1d is retained, a no antibiotic control should be included to compare the results in Fig 1D showing the synergistic effects of INH and BDQ. Additionally, 2.7 ug/ml was chosen as the baseline concentration to investigate the mechanisms of BDQ sensitivity across the study, but a reason for this was not stated in the manuscript.

As stated above, we have removed this laboratory evolution experiment from our revised manuscript.

We initially chose 2.7 µg/mL BDQ because this concentration conferred the strongest killing phenotype in our time-kill experiments combining INH and BDQ (below). From these time-kill CFU data and the CellROX ROS measurements in Figure 2e, we think we would have found very similar results if we used 0.68 µg/mL BDQ throughout this project.

Likewise, 5.4 ug/ml of BDQ was used to demonstrate reactive oxygen species formation in KatG-deficient mutants (Fig 2/ lines 144-168) but the reasoning behind this is not clearly stated.

Following concerns raised by Reviewers 2 and 3 on the BDQ concentrations used in our experiments, we carefully reviewed all of the laboratory notebooks for this project. We found that the 5.4 µg/mL concentration stated for the 8-oxo-dG experiments was incorrectly stated in the manuscript. The reason for this mistake was because in the technical execution of the ELISA experiments, 5 mL of mid-log Mtb culture was added to 5 mL 5.4 µg/mL BDQ to achieve 10 mL Mtb cultures with 2.7 µg/mL BDQ. We carefully corrected typographical errors for the concentrations used in our ELISA experiments in the figure legends for Figure 2e, 2f and 2g and in the methods sections.

The authors should mention that the E553K mutations (line 112) are probably a result of picking sibling colonies. They should also quantify the level of resistance of their katG-E553K strain, and note its relevance to clinical resistance.

As stated above, we have removed the experiments involving KatG_{E553K} from our revised manuscript.

The full genotype of auxotrophic strains mc28245 and mc27902 should be mentioned in the main text. Would the auxotrophy be expected to change the metabolism and affect the results?

We have added genotypes for the mc²8245 and mc²7902 to the main text as follows:

Lines 142-145: "To test this hypothesis, we measured BDQ-induced ROS accumulation in the INH-resistant BSL-2 auxotrophic strain mc²8245, which harbors a large genomic deletion spanning over

katG (H37Rv Δ panCD Δ leuCD Δ argB Δ 2116169-2162530), and its INH-susceptible ancestor mc²7902 (H37Rv Δ panCD Δ leuCD Δ argB)²⁵.”

Analysis and interpretation of dose response curves requires more discussion, especially where they do not strongly support the claims in the text.

For example, the differences observed in the MIC BacTiter-Glo and the OD600 results in Fig 3C& D is not discussed, but the authors conclude (lines 188-190) that overexpression of *kmtR* and *Rv3160c* sensitized the strains to BDQ. In particular, for OD600 measurement of *kmtR*, there appears to be little if any sensitization.

The BacTiter-Glo assay depends on cellular ATP, but is used to measure viability in the presence of the ATP synthase inhibitor BDQ. How would that influence the observed results?

We wish to clarify that the experiments that Reviewer 2 refers to were not BDQ lethality experiments and therefore not measurements of loss of viability. These experiments were instead growth experiments from a low starting cell density. We think that BacTiter-Glo and turbidity (OD₆₀₀) report on two different aspects of Mtb physiology: ATP production vs growth. In the case of *Rv3160c*, the BDQ sensitivity of both ATP production and growth are higher in *Rv3160c* overexpressing cells than in control cells (Figure 3b). In the case of *kmtR*, only ATP production is sensitized to BDQ in *kmtR* over-expressing cells (Figure 3c). In the case of *prpR*, there is no significant difference between *prpR* and control cells (Supplementary Fig. 3c). We have clarified these in the text as follows:

Lines 194-202: “We measured the BDQ sensitivity for inhibition of ATP synthesis by BacTiter-Glo and BDQ sensitivity for growth inhibition by turbidity (OD₆₀₀). In support of our hypothesis, BDQ-mediated inhibition of ATP synthesis in cells over-expressing *Rv3160c* or *kmtR* was sensitized to BDQ (Fig. 3b and 3c). In addition, *Rv3160c* over-expression sensitized cells to BDQ-mediated growth inhibition. However, we did not observe changes in BDQ sensitivity for either ATP production or growth in *prpR* over-expressing cells (Supplementary Fig. 3c). Nonetheless, the sensitized BDQ-mediated inhibition of ATP synthesis and growth in *Rv3160c* over-expressing cells suggest *Rv3160c* and other transcriptional programs induced by deficient catalase activity may also contribute to BDQ hypersusceptibility in drug-resistant Mtb.”

Dose response curves should also be normalized to show percentage growth (or inhibition). For example, Fig. 1g may not show increased susceptibility if normalized.

As stated above, we have removed experiments involving KatG_{E553K} from our revised manuscript. For most experiments, we chose to not normalize dose response curves either because the uninhibited measurements (at very low drug concentrations) were similar between control and mutant strains or because growth was not fully uninhibited at the lowest concentrations. We also think the less manipulated raw measurements are a more complete representation of the results from our experiments.

However, we have normalized measurements from experiments involving the clinical strains because these strains exhibit meaningful growth variation, as Reviewer 3 suggests (Figures 2d, 4d, 5b, and 5c). We have moved the non-normalized data for these experiments to the supplement (Supplementary Fig.2a, 4b, and 6c).

In the right-hand panel of Fig. 5b, there is not susceptibility of the mutant to SMX-TMP. The authors should discuss why that might be.

We think the reason why $\Delta katG$ does not have increased susceptibility to SMX-TMP over WT cells is because WT cells are already very susceptible to SMX-TMP when compared with SMX or TMP susceptibility alone. We think it is likely that the susceptibility effect is already saturated by for both WT and $\Delta katG$ cells at these concentrations in our experiment. We have added this explanation to our revised manuscript as follows:

Lines 287-290: "Moreover, while $\Delta katG$ and wild-type cells were both sensitized to the combination of TMP and SMX (Supplementary Fig. 6d) relative to TMP alone (Fig. 5b), $\Delta katG$ cells did not exhibit any further TMP-SMX sensitization than wild-type cells, possibly due to saturation of the growth inhibition that can be achieved by folate biosynthesis inhibition."

Minor comments

The results section would flow better if the figures were arranged in a similar order and numbered accurately. For example, in lines 89-172, the figures (Fig.1) cited do not currently correspond to the figures they refer to.

We have carefully corrected all errors on figure references throughout the manuscript. Some figures have changed and all figure references in the text should now be correct.

What is the breakpoint for INH resistance used in this study?

The CRyPTIC Consortium defines the INH resistance breakpoint as 0.8 $\mu\text{g}/\text{mL}$ and classified strains as INH-susceptible if their INH MIC was $\leq 0.1 \mu\text{g}/\text{mL}$ (CRyPTIC Consortium, PLoS Biology 2022). Each of the INH^R strains used in this study possessed INH MICs $\geq 3.2 \mu\text{g}/\text{mL}$, as measured by the TDR-TB strain bank (Vincent, Int J Tuberc Lung Dis 2012). Each of these INH^S strains possessed INH MICs $\leq 0.2 \mu\text{g}/\text{mL}$, as measured by the TDR-TB strain bank. We have added this detail in the text as follows:

Lines 87-89: "Each INH-susceptible strain possessed INH MICs $\leq 0.2 \mu\text{g}/\text{mL}$ and each INH-resistant strain possessed INH MICs $\geq 3.2 \mu\text{g}/\text{mL}$, as reported by the TDR-TB strain bank."

Methodology should be mentioned in figure legends or the text where they are referenced. For example, it is not clear how gene expression in Fig. 2b, 3a, 3b, 4a etc are measured.

We have updated figure legends to indicate that gene expression measurements come from the RNA-seq experiments. We have carefully revised figure legends to more explicitly specify experimental methods.

Fig 3- legend needs to be edited as the captions do not match.

We have corrected Figure 3 to match the caption.

The scales on Fig 4 should be edited to match each other for clarity and comparison

The scales for the dose-response growth inhibition experiments in Figure 4 are matched, while the RNA sequencing gene expression measurements are not. We do not think the expression of different genes should be compared with each other as some genes because the basal expression of genes is widely varying. For example, Rv0440 (*groEL*) in our RNA sequencing data has a \log_2 normalized expression of ~ 17.1 in untreated WT H37Rv cells, while the median \log_2 normalized expression for all genes was 10.1 (> 100 -fold increase in baseline expression). In the case of the gene expression counts depicted in Figure 4a, baseline expression for *alkA*, *radA*, *recG*, and *ung* ranged from ~ 10 to ~ 12 and their group maximum and minimum expression was ~ 13 and ~ 9.25 , respectively. We choose to not match scales for the gene expression counts in Fig. 4a so that the change in expression would not be compressed when visualizing gene expression changes in these data.

The discussion section could also benefit from having the figures mentioned where they are discussed. They are referred to in a few instances.

We have now added figure references throughout the discussion sections to cite where data generated in this work can be found.

The authors correctly note that TMP and SMX are not used in TB treatment, but should note that the antifolate PAS was part of the first combination therapy for TB. It would strengthen the manuscript to repeat the experiments of Fig. 5 with PAS, but it is not essential.

We have added information and discussion on PAS to the discussion section as follows:

Lines 379-383: “While TMP and SMX are widely used as bacteriostatic antibiotics against several bacterial pathogens, they are not used in treating TB. Instead, the antifolate para-aminosalicylic acid (PAS) has been used as a second-line antibiotic for treating MDR-TB^{49,50}. However, clinical utilization of PAS for treating (MDR-)TB is limited by its high toxicity⁵¹. Our data suggest folate biosynthesis may still be useful as a therapeutic target for curing TB.”

Remarks on code availability: The repository only contains a README file, but no code. Presumably the authors will update the repository in due course.

Code for the RNA sequencing analyses and genome-scale metabolic modeling is now deposited in the following GitHub repository: https://github.com/jasonhyang/OforiAnyinam_2024 . Raw sequencing data has now been deposited in the following Sequence Read Archive BioProject: PRJNA1139169.

Reviewer 4

We are happy Reviewer 4 received an opportunity to participate in peer review and appreciate their comments above.

Reviewer 5

We are happy Reviewer 5 received an opportunity to participate in peer review and appreciate their comments above.